# TGEA 2.0: A Large-Scale Diagnostically Annotated Dataset with Benchmark Tasks for Text Generation of Pretrained Language Models

**Huibin Ge**[†], **Xiaohu Zhao**[†], **Chuang Liu**[†],
**Yulong Zeng**[§], **Qun Liu**[§] and **Deyi Xiong**[†]
[†] College of Intelligence and Computing, Tianjin University, Tianjin, China
[§] Huawei Noah's Ark Lab, Hong Kong, China
{gehuibin, zhaoxiaohu, liuc_09, dyxiong}@tju.edu.cn
{zengyulong, qun.liu}@huawei.com

## Abstract

In order to diagnostically analyze and improve the capability of pretrained language models (PLMs) in text generation, we propose TGEA 2.0, to date the largest dataset built on machine-authored texts by PLMs with fine-grained semantic annotations on a wide variety of pathological generation errors. We collect 170K nominal, phrasal and sentential prompts from 6M natural sentences in 3 domains. These prompts are fed into 4 generative PLMs with their best decoding strategy to generate paragraphs. 195,629 sentences are extracted from these generated paragraphs for manual annotation, where 36K erroneous sentences are detected, 42K erroneous spans are located and categorized into an error type defined in a two-level error taxonomy. We define a **Mi**nimal **S**et of **E**rror-related **W**ords (MiSEW) for each erroneous span, which not only provides error-associated words but also rationalizes the reasoning behind the error. Quality control with a pre-annotation and feedback loop is performed before and during the entire annotation process. With the diagnostically annotated dataset, we propose 5 diagnosis benchmark tasks (i.e., erroneous text detection, MiSEW extraction, erroneous span location and correction together with error type classification) and 2 pathology mitigation benchmark tasks (pairwise comparison and word prediction). Experiment results on these benchmark tasks demonstrate that TGEA 2.0 is a challenging dataset that could facilitate further research on automatic diagnosis and pathology mitigation over machine texts. The dataset is publicly available at https://github.com/tjunlp-lab/TGEA/.

## 1 Introduction

Pretrained language models [30, 37, 33, 33, 5], which learn general knowledge from enormous amounts of data, achieve impressive results in text generation [1, 33, 14]. Given a textual prompt (e.g., a word, phrase or an utterance, instruction), generative PLMs are capable of producing subsequent texts that either cohere with the prompt or complete the task indicated in the prompt. Despite the substantial success on this aspect, a variety of issues have been found in PLM-authored texts, with respect to privacy [3], toxicity [24], quality, etc. The key interest in this paper is the quality (e.g., grammatical correctness, semantic coherence) of machine texts generated by PLMs.

A series of datasets and benchmarks have been curated for human-written texts [22, 23, 2, 21, 9]. In contrast, resources related to PLM-authored texts are quite limited although plenty of holistic/task-specific manual/automatic evaluations have been done in text generation [4]. To the best of our

---

Corresponding author: Deyi Xiong (dyxiong@tju.edu.cn).

knowledge, TGEA [12] and SCARECROW [7] are two recently proposed benchmark datasets that aim to scrutinize and taxonomize errors in PLM-authored texts in Mandarin Chinese and English, respectively. Unfortunately, the scale of the two datasets, in terms of the number of sentences/paragraphs (47K/1.3K) annotated, is small in comparison to the size of datasets built on human-written texts, e.g., CGEC dataset [40] that consists of 717K sentences.

From the aforementioned datasets with manual annotations over machine-generated texts, we can gain an insight into the patterns that PLMs produce errors in text generation and the distribution of these error patterns and types. With sufficient annotations, we could even learn a diagnostor to automatically detect errors in texts authored by PLMs and categorize these errors into predfined types [12, 7]. More interestingly, an important goal of annotating errors in PLM-generated texts is to improve the capability of PLMs in text generation by teaching them to avoid producing pathological texts with annotated instances in a variety of ways (e.g., contrastive learning, error-driven learning). Unfortunately, this has not yet been explored in the context of PLM-authored generation.

In this paper, we propose a large-scale diagnostically annotated dataset with a fine-grained error ontology, TGEA 2.0, aiming to (1) reveal a spectrum of pathologies in PLM-authored generation, (2) enable automatic diagnosis on erroneous span location and classification and (3) teach PLMs to mitigate pathological generation. We collect machine-generated texts from multiple sources with different model architectures and configurations. The contributions of TGEA 2.0 are as follows.

- **Scale and Diversity.** TGEA 2.0 is to date the largest error-annotated dataset built on PLM-generated texts. It contains 195,629 diagnostically annotated long sentences generated by different PLMs, including NEZHA-Gen [36], GPT-2 [27], PanGu-$\alpha$ [37] and CPM [39], with different model scales (the number of parameters ranging from 110M to 2.6B), neural architectures and decoding strategies. Natural prompts that are used to trigger text generation are collected from 3 domains (i.e., News, Wikis, Fictions) and in 3 different schemes (i.e., nouns, phrases and sentences).

- **Semantic Annotations.** Machine texts in TGEA 2.0 are manually annotated with rich information, including erroneous text indicator, erroneous text spans, error types (from a two-level error taxonomy with 24 errors [12]), corrections to errors, etc. Particularly, we define a **Mi**nimal **S**et of **E**rror-related **W**ords (MiSEW) for each erroneous span, which can not only provide rich erroneous information with fewer meaningless redundant words, but also be regarded as a simplifed explanation to justify the corresponding error annotation.

- **Analysis.** We conduct an in-depth analysis on the annotated dataset, showing pathological errors, their types and distributions. We find that different error patterns are exhibited across different models and different datasets (TGEA 2.0 vs. TGEA/SCARECROW).

- **Benchmarking.** TGEA 2.0 provides two groups of benchmarking tasks: diagnosis tasks and pathology mitigation tasks. The former consist of erroneous text detection, erroneous span location, error type classification, MiSEW extraction and error correction. PLMs, e.g., RoBERTa, BART, are used in these diagnosis tasks. The best performance across all diagnosis tasks achieved by these strong models is $< 54\%$ in $F_1$, suggesting that TGEA 2.0 is challenging. In the pathology mitigation tasks, we propose MF-GPT-2 (-w BC), a Chinese GPT-2 model trained on TGEA 2.0 with both generative and discriminative fine-tuning. It gains improvements of 18.93 and 3.62 points in accuracy in pairwise comparison and word prediction respectively, demonstrating that TGEA 2.0 is useful for improving text generation capability of PLMs.

## 2 Related Work

Our dataset is closely related to TGEA [12] and SCARECROW [7], which have been recently proposed. TGEA annotates errors on 47K machine texts generated by a Chinese GPT-style PLM NEZHA-Gen [36] with an error taxonomy covering 24 error types. SCARECROW is an English dataset that has 13K manual annotations over 1.3K human-written or machine-generated paragraphs with an error ontology of 10 error types. Significant differences between TGEA 2.0 and SCARE-CROW lie not only in language (Chinese vs. English) and dataset size (195K vs. 13K), but also in the following aspects. First, all texts in TGEA 2.0 are machine-authored, containing correct and erroneous texts judged by annotators. Second, the prompts used in TGEA 2.0 are more diversified than those in SCARECROW in terms of both domain and type. Third, we follow a very different

annotation procedure to guarantee annotation quality and use a different error taxonomy that is tailored for error patterns in our data.

There are four key differences between TGEA 2.0 and TGEA 1.0 while the scale difference is only one of them. First, partially inspired by SCARECROW, we employ multiple generative language models with different configurations and decoding strategies to further diversify generated texts, which is significantly different from TGEA that uses only one pretrained language model for generation. The second substantial extension to TGEA is that we define a minimal set of error-related words for each erroneous span to substitute rationals written by annotators in TGEA. See Section 3.2 for more details on this. Third, we perform error type classification and error correction based on ground-truth erroneous spans to measure the degree of difficulties of these tasks as standalone tasks. Regarding benchmarking tasks over the dataset, TGEA 2.0 not only offers a variety of diagnosis tasks (e.g., error detection, classification, MiSEW extraction), but also a new task, termed as "Pathology Mitigation", for improving the generation capability of PLMs with the annotated data, which is not available in both TGEA and SCARECROW.

TGEA 2.0 is also partially related to commonsense reasoning datasets on PLMs, usually in the form of QA with multi-choice questions. To name a few, COPA [28] aims at commonsense causal reasoning, finding causes or effects for given premises. WinoWhy [38] requires models to choose plausible reasons for Winograd Schema Challenge questions. These datasets attempt to explicitly probe the commonsense reasoning capability of PLMs by forcing them to answer questions, instead of evaluating texts produced by PLMs. SemVE [34] evaluates whether a system can distinguish a natural language statement that makes sense and provide the reasons. It treats reasoning as a generation task which is similar to TGEA. Due to the difficulty of generating plausible rationales, TGEA 2.0 uses MiSEWs to avoid bias introduced by annotators, rather than human-written explanations, for reasoning behind error annotation.

## 3 TGEA 2.0 Dataset

### 3.1 Data Collection

We collect machine texts from the combination of three aspects: model, decoding strategy and prompt. The motivation behind this is to obtain diversified erroneous data from different models under their optimal settings.

**Model.** We select four GPT-like generative language models that are publicly available to generate continuations to the given prompts. These models are NEZHA-Gen [36] with 110M parameters, GPT-2 [27] with 1.5B parameters, PanGu-$\alpha$ [37] with 2.6B parameters and CPM [39] with 2.6B parameters. The variations in the architecture and scale of these models may allow us to collect machine texts that exhibit different error patterns and distributions.

**Decoding Strategy.** We want each selected PLM to generate the continuation to a given prompt under its best generation strategy and setting. Texts generated with conventional decoding strategies, like beam search, greedy search, usually contain plenty of repetitions, resulting in poor readability. Hence we use stochastic decoding methods: Top-$p$ sampling (i.e., nucleus sampling), Top-$k$ sampling and sampling with temperature [13]. To reduce repetition, for all strategies we set the repetition penalty to 1.2. We start our initial study to compare two decoding strategies: Top-$k$ sampling ($k$=30, $t$=0.9) and Top-$p$ sampling ($p$=0.9, $t$=0.9). We find they achieve similar generation quality. Since the Top-$p$ sampling is able to capture the vocabulary usage in a distribution close to the human distribution [13], we vary $p$ and $t$ to select the best setting for each of the four target PLMs.

**Prompt.** We have randomly sampled 6M sentences from 3 different domains, i.e., News, Wikipedia, Web Fictions (2M sentences per domain). These natural sentences from real-world texts are then used as a data pool, from which three types of prompts are extracted:

- **Nominal Prompt.** Nouns with an occurrence frequency $f \in [20, 200]$ in the data pool are extracted as nominal prompt candidates. These nouns are neither rare words nor sight words, ensuring that continuations to them generated by PLMs are fluent and meaningful.

- **Phrasal Prompt.** Phrases, separated by commas, containing at least one word from the top 30% high-frequency word list in the data pool are extracted.

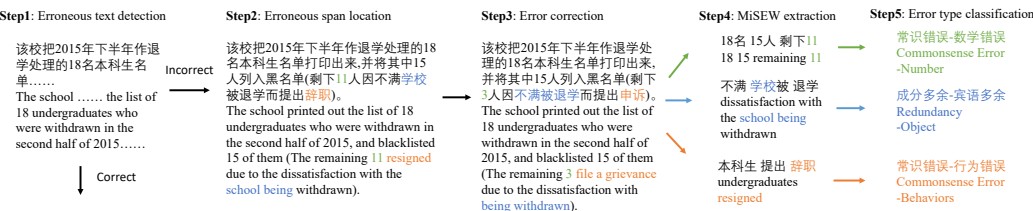

Figure 1: Illustration of data annotation.

- **Sentential Prompt.** Similarly, sentences containing at least one word from the list of top 30% frequent words are extracted as candidates. Please note that sentential prompts are removed after continuations to them are generated by PLMs so as to reduce the workload of manual annotation.

Our final prompts are a mixture of these three types of prompts. In order to set the proportion for each prompt type in the mixture, we randomly choose 40 prompts for each type and feed them to each of the four target PLMs. We ask annotators to judge whether the continuations to these prompts are correct. Statistics are shown in Table 1 in Appendix A.1, which demonstrate that phrasal prompts are more likely to yield erroneous continuations. Thus we select prompts from the mixture of nominal, phrasal and sentential prompts in a proportion of 3:5:2 to encourage erroneous texts to be generated as many as possible.

In order to find the best decoding setting for each target PLM, we continue to use these randomly selected prompts to feed into them with varying $p$ and $t$. The numbers of erroneous texts generated by the four PLMs are shown in Table 2 in Appendix A.1. The decoding setting with the smallest number of erroneous texts is then used for the corresponding model.

We collect 170,000 prompts, among which 10,000 are shared by the four PLMs while the remaining 160,000 are divided equally across the four models. In this way, we obtain 200K paragraphs, where sentences with noisy fragments, e.g., [UNK], URLs, and sentences that contain more than 30% non-Chinese characters, are removed. From the remaining sentences of each generated paragraph, we select the first sentence as the final sentence for annotation. The reason for not annotating the entire paragraphs is that they are very long and hence it is very challenging for annotators to assess. On the other hand, the selected Chinese sentences are composed of a sufficient number of tokens (averagely 44.36 characters in our dataset). These sentences are usually containing heavy content that could be expressed in multiple sentences [16], which are hence linguistically expressive for error annotation.

## 3.2 Data Annotation

The annotation procedure is illustrated in Figure 1. There are 5 stages of annotation: erroneous text detection, erroneous span location, error correction, MiSEW extraction, error type classification.

**Erroneous Text Detection.** The first stage of annotation is to detect erroneous texts from given machine texts. Although we do not include the context of a given text from its original paragraph generated by PLMs, for the majority of collected machine texts, the correctness of them can be judged according to the given texts themselves. For those that cannot be judged by themselves, we allow annotators to "imagine" their context and make an assessment. In doing so, we attempt to keep false positives of erroneous texts to a minimum. The detected erroneous texts are then passed to the subsequent stages for further annotation.

**Erroneous Span Location and Correction.** The next two tasks for annotators are to locate erroneous spans from erroneous texts and correct them. Like TGEA [12], we follow the minimal edit principle to correct errors and discard texts that annotators think it difficult to correct with this principle. All located erroneous spans in each given text are required to be corrected.

**MiSEW Extraction.** In addition to erroneous span location and correction, we define a **Mi**nimal **S**et of **E**rror-related **W**ords (MiSEW) for each erroneous span, which are either discrete words or continuous spans. MiSEWs are in line with the following principles:

- With the minimal set of error-related words, humans can detect the error and propose a reasonable correction without reading the original sentence.

- Deleting any items in MiSEW makes it impossible for annotators to detect the error.

- The erroneous span is contained in MiSEW.

The reason for extracting a MiSEW for each erroneous span is three-fold. First, a MiSEW is self-evident, which can be regarded as an explanation of annotators for their reasoning behind the corresponding error. Second, extracting MiSEWs reduces the annotation time in comparison to writing a long rationale. Third, as MiSEWs are extracted from machine texts, they do not introduce annotators' bias.

**Error Type Classification.** Inspired by TGEA [12], we also categorize the annotated errors into error types with 5 level-1 error types and 24 level-2 error types.

**Annotation Confidence.** In addition to the aforementioned annotations, annotators are also required to provide a confidence score (very confident, confident and unconfident) for each annotated sentence.

### 3.3 Quality Control

Partially inspired by previous dataset annotation [12, 19, 26, 35], our quality control procedure starts with a pre-annotation stage, where annotators are trained to deeply understand the annotation convention for the dataset. In the first step of this stage, two reviewers from TGEA 1.0 [12] with an average IAA of 92.3% and Cohen's Kappa of 82.6% train four reviewers. The trained four reviewers then annotate 1000 texts. These annotations are used as golden-standard results to train the annotators in three times. As shown in Table 3 in Appendix A.1, we gradually expand the pre-annotation data from 50 sentences to 800 sentences. Each time the annotators return their results, the four reviewers carefully review these results and provide instant feedback on common annotation issues to the annotators. In this feedback loop, the average performance of the annotators increases from 58.9% to 79.7% across different annotation tasks. Given the inter-annotator disagreement and difficulty of the annotation tasks, the final performance in the pre-annotation stage suggests that the annotators are well trained for annotation.

During the official annotation stage, we continue to monitor annotation quality by both automatic and manual check & review. Automatic check uses rules to detect annotations that violate basic annotation principles (e.g., "erroneous span must be contained in MiSEW"). After automatic check, 7 well-trained reviewers with an average Cohen's Kappa of 56.6% manually check annotation results delivered in batches from our subcontractor. Problematic annotations found in automatic and manual review are sent back to the corresponding annotators for re-annotation. Particularly, as the currently used four PLMs are from public sources, there are a very few generated sentences containing Yue (Cantonese) characters. During the annotation, if a sentence with Yue characters is unreadable for Mandarin Chinese-speaking annotators, such sentences are removed (A detailed description on the variety of Chinese in the training data of the four PLMs is provided in Appendix A.5).

## 4 Dataset Analysis

### 4.1 Overall Statistics

After annotation, we obtain 195,629 annotated sentences. We reshuffle these sentences and divide them into the training/dev/test set according to a proportion of 8:1:1. As shown in Table 1, TGEA 2.0 contains 36,023 erroneous sentences with 42,067 erroneous spans that are annotated with semantic information. We also display the distribution of erroneous sentences according to the number of erroneous spans contained in them. Sentences with 1/2/3 erroneous spans account for 86.33/11.20/1.99% of all erroneous sentences.

On average, a MiSEW contains 10.93 characters, which accounts for 24.64% of characters in erroneous sentences. We also provide the statistics on the confidence of the annotators. Among all erroneous sentences, 62.20% sentences are annotated with a confidence score of "very confident", 35.41% texts with "confident" and 2.39% with "unconfident". This suggests that the overwhelming majority of erroneous sentences (97.61%) are labeled by annotators confidently.

|         | Train | Dev | Test | All |
|---------|-------|-----|------|-----|
| #S | 156,502 | 19,563 | 19,564 | 195,629 |
| cS | 127,684 | 15,961 | 15,961 | 159,606 |
| w/ 1 es | 24,862 | 3,129 | 3,106 | 31,097 |
| w/ 2 es | 3,249 | 385 | 401 | 4,035 |
| w/ 3 es | 567 | 70 | 80 | 717 |
| w/>= 4 es | 140 | 18 | 16 | 174 |
| T.eS/s | 28,818/33,666 | 3,602/4,181 | 3,603/4,220 | 36,023/42,067 |
| Avg.c.S/MiSEW | 44.41/10.92 | 44.34/11.08 | 44.36/10.87 | 44,36/10.93 |
| Avg.t.MiSEW | 3.40 | 3.39 | 3.38 | 3.40 |
| VC/C/UC | 62.32/35.22/2.46 | 61.66/36.37/1.97 | 61.84/35.94/2.22 | 62.20/35.41/2.39 |

Table 1: Data statistics of TGEA 2.0. S: sentences. cS: correct sentence. es: erroneous spans. T.eS/s: Total erroneous sentences/spans. Avg.c.S/MiSEW: the average number of characters in erroneous sentences/MiSEWs. Avg.t.MiSEW: the average number of tokens segmented by annotators in MiSEWs. VC/C/UC: Very confident/Confident/Unconfident (%).

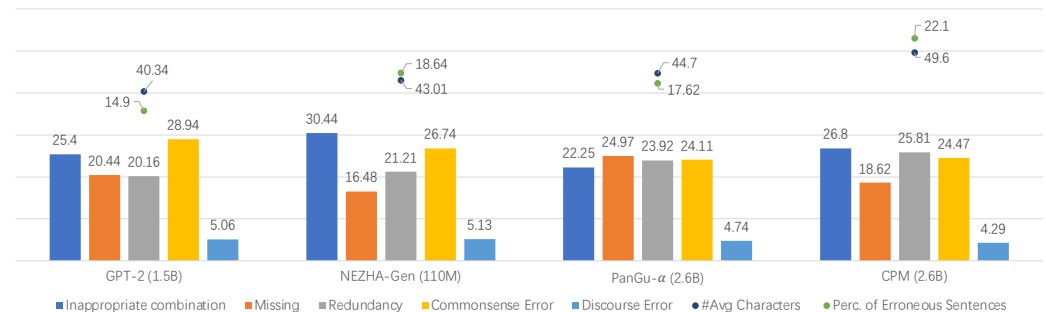

Figure 2: The percentage of erroneous sentences and the average number of characters in erroneous sentences generated by the four PLMs with error distributions over 5 error types (%).

## 4.2 Error Distributions of PLMs

We show the percentage of erroneous sentences and average number of characters of sentences generated by 4 different Chinese generative PLMs in Figure 2. The percentage of erroneous sentences produced by CPM with 2.6B parameters is 22.10%, the highest among all models. It seems that this percentage is not highly related to the number of parameters in models, but sensitive to the length of sentence generated, suggesting that the longer sentences generated by PLMs are, the more likely PLMs make mistakes.

## 4.3 Percentage of Erroneous Sentences

Although TGEA 2.0/TGEA and SCARECROW use different error taxonomies on different languages, we compare the distributions over error types of these three datasets in Figure 1 in Appendix A.1 to gain insights into common problems in text generation exhibited by generative PLMs. Clearly, grammatical errors are still the major issues in PLM-based text generation, accounting for 64.87% (e.g., inappropriate combination, redundancy, missing) in TGEA, 36.75% (e.g., grammar usage, redundant) in SCARECROW and 69.23% (e.g., inappropriate combination, redundancy, missing) in TGEA 2.0. The commonsense errors are also frequent common errors among the three datasets.[1] Compared with TGEA, TGEA 2.0 has a significantly lower percentage of redundancy errors due to the stochastic decoding strategy and repetition penalty used.

Additionally, we further analyse the distributions over error types for different PLMs, which are illustrated in Figure 2. Redundancy errors tend to appear more frequently in longer sentences (e.g.,

---

[1]Note that error types in SCARECROW, e.g., commonsense, bad math, are roughly corresponding to the commonsense error type in the error taxonomy of TGEA/TGEA 2.0, which is further divided into errors like number, time errors at the level 2.

sentences generated by PanGu-$\alpha$ and CPM). It is also interesting to find that commonsense errors are more likely to occur in small models.

## 5  Benchmark Experiments

We use TGEA 2.0 as a benchmark testbed for both diagnosis tasks and pathology mitigation tasks. The former include erroneous sentence detection, MiSEW extraction, erroneous span location, error type classification, error correction while the latter offer two subtasks: pairwise comparison and word prediction.

We employ two BERT-style Chinese PLMs, RoBERTa-wwm-ext and MacBERT-base [6] and two Chinese generative PLMs, GPT2-cluecorpussmall [8] and Chinese BART-base [29] as baseline models in our experiments. We denote them as RoBERTa, MacBERT, GPT-2 and BART for notational simiplicity. Detailed experiment settings are provided in Appendix A.2.

### 5.1  Erroneous Sentence Detection

**Task Definition.**  This is the same as defined in TGEA [12], which is to automatically identify whether a given machine text is erroneous.

**Model.**  We treat this task as a text classification task and fine-tune RoBERTa and MacBERT on our training data. As the data are imbalanced in which correct texts account for 81.58%, we employ focal loss and $\alpha$-balanced loss [18] to alleviate this problem.

**Results.**  As shown in Table 2, all models are not able to significantly outperform "Majority" (i.e., predicting all as correct instances) in accuracy and "Minority" (i.e. predicting all as incorrect instances) in $F_1$. Even with the focal loss and $\alpha$-balanced loss that attempt to mitigate the data imbalance issue (see Appendix A.3), models still perform poorly, with accuracy scores lower than 82% and $F_1$ scores lower than 32%, suggesting that this is a very challenging text classification task.

### 5.2  MiSEW Extraction

**Task Definition.**  This is a task that automatically predicts the minimal set of error-related words given an erroneous text. This can be done as sequence labeling by regarding words in MiSEW as positive words and other words in the erroneous text as negative words.

**Model.**  Sequence labeling style task-specific models are formed by incorporating the Chinese pretrained language models (e.g., MacBERT) and an additional output layer that predicts labels for words. These models are then fine-tuned on the training data. To alleviate the data imbalance issue, $\alpha$-balanced loss is also used.

**Results.**  As shown in Table 2, the best model achieves 72.72%/72.91% in accuracy and 39.32%/38.76% in $F_1$ on the development/test set. The performance in $F_1$ is even lower than Minority (43.96%/43.63%). Using the $\alpha$-balanced loss obtains a gain of 11.63%/11.80% and 11.57/11.22% in $F_1$ for RoBERTa and MacBERT on the development/test set, respectively.

### 5.3  Erroneous Span Location

**Task Definition.**  As MiSEWs are closely related to erroneous spans and erroneous spans are included in MiSEWs, we define the detection of erroneous spans over MiSEWs. That is, given the ground-truth/extracted MiSEWs, we further pinpoint erroneous spans in them in a pipeline fashion. Alternatively, the erroneous span location and MiSEW extraction can be also defined as a joint sequence labeling task where erroneous spans and MiSEWs are automatically recognized at the same time or as two separate tasks that are performed independently. Here we only discuss models and results for the pipeline method (ground-truth MiSEWs are used) while results of joint and separate recognition are reported in Appendix A.3.

**Models.**  Similar to MiSEW extraction, erroneous span location can be performed as a sequence labeling task by regarding words in erroneous spans as positive words, and other words in MiSEWs as negative words. As a MiSEW is a set of words which may not match the input of PLMs, we first extract representations of MiSEWs from PLMs. Specifically, we feed the complete erroneous text

| Task | Model | Dev | | | | Test | | | |
|---|---|---|---|---|---|---|---|---|---|
| | | A (%) | P (%) | R (%) | $F_1$ (%) | A (%) | P (%) | R (%) | $F_1$ (%) |
| Erroneous Text Detection | Majority | 81.59 | 0 | 0 | 0 | 81.58 | 0 | 0 | 0 |
| | Minority | 18.41 | 18.41 | **100.0** | 31.10 | 18.42 | 18.42 | **100.0** | **31.10** |
| | RoBERTa | 81.84 | 55.17 | 7.41 | 13.07 | 81.99 | 58.35 | 7.66 | 13.54 |
| | RoBERTa-$\alpha$=0.3 | 80.48 | 44.09 | 22.35 | 29.66 | 80.50 | 44.08 | 21.93 | 29.29 |
| | MacBERT | **81.92** | **57.14** | 7.22 | 12.82 | **82.07** | **59.96** | 7.94 | 14.02 |
| | MacBERT-$\alpha$=0.3 | 80.07 | 42.8 | 24.43 | **31.11** | 80.35 | 43.83 | 23.84 | 30.88 |
| | Human | 91.13 | 76.19 | 72.72 | 74.42 | 91.39 | 73.08 | 76.00 | 74.51 |
| MiSEW Extraction | Majority | 69.30 | 0.00 | 0.00 | 0.00 | 69.55 | 0.00 | 0.00 | 0.00 |
| | Minority | 30.70 | 30.70 | **100.0** | **43.96** | 30.45 | 30.45 | **100.0** | **43.63** |
| | RoBERTa | 72.64 | 41.08 | 25.55 | 27.69 | 72.79 | 39.66 | 25.00 | 26.96 |
| | RoBERTa-$\alpha$=0.4 | 71.69 | 44.38 | 43.21 | 39.32 | 71.67 | 44.12 | 42.94 | 38.76 |
| | MacBERT | **72.72** | 40.08 | 25.14 | 27.25 | **72.91** | 39.72 | 25.02 | 27.00 |
| | MacBERT-$\alpha$=0.4 | 72.00 | **45.21** | 41.88 | 38.82 | 72.02 | **44.75** | 41.66 | 38.22 |
| Erroneous Span Location | Majority | 75.36 | 0.00 | 0.00 | 0.00 | 75.41 | 0.0 | 0.0 | 0.0 |
| | Minority | 24.64 | 24.64 | **100.0** | **36.70** | 24.59 | 24.59 | **100.0** | **36.67** |
| | RoBERTa | 76.78 | 21.39 | 14.34 | 15.28 | **76.70** | 21.14 | 15.15 | 15.86 |
| | RoBERTa-$\alpha$=0.35 | 75.41 | 33.95 | 33.44 | 29.80 | 75.57 | **34.37** | 35.49 | 31.08 |
| | MacBERT | **76.82** | 23.36 | 16.69 | 17.37 | 76.60 | 22.85 | 17.02 | 17.50 |
| | MacBERT-$\alpha$=0.35 | 75.19 | **34.50** | 34.93 | 30.72 | 75.11 | 34.00 | 36.50 | 31.32 |
| Error Type Classification | RoBERTa | **52.43** | **53.89** | 51.97 | **52.57** | 52.70 | **55.35** | 52.21 | **53.20** |
| | MacBERT | 52.12 | 52.74 | **52.51** | 52.56 | **53.11** | 53.39 | **53.05** | 53.11 |
| | | Precision | Recall | $F_{0.5}$ | | Precision | Recall | $F_{0.5}$ | |
| Error Correction | BERT-GEC | 0.89 | 2.39 | 1.01 | | 1.17 | 3.10 | 1.33 | |
| | BART | **13.39** | **31.46** | **15.12** | | **14.17** | **33.15** | **16.01** | |

Table 2: Results on the five diagnosis tasks. A: Accuracy. P: Precision. R: Recall.

into the model and get the representations of all words. We then mask the words that do not belong to MiSEWs and perform sequence labeling over the remaining words.

**Results.** The best model with the $\alpha$-balanced loss achieves 76.82% in accuracy and 30.72% in $F_1$ on the development set and 76.70% in accuracy and 31.32% in $F_1$ on the test set, which still underperforms Minority (36.70%/36.67%) on the development/test set.

## 5.4 Error Type Classification

**Task Definition.** This is a text classification task on erroneous spans. We only perform level-1 error type classification in the form of 5-way classification. In order to measure the degree of difficulty of this task as a standalone task, we use ground-truth erroneous spans during the training and inference of error type classification. In this way, the pipeline impact of erroneous span location on the error type classification is not considered.

**Models.** As an erroneous text may contain multiple erroneous spans, we perform this task in a straightforward way: categorizing each erroneous span separately and independently into the corresponding level-1 error type. An error indicator embedding, which distinguishes words in an erroneous span from other words, is added to the input of the fine-tuned models. Additional results are provided in Appendix A.3.

**Result.** Macro-precision/recall/$F_1$ results are reported. Again, the results are very low, the best model achieves 52.43%/53.11% in accuracy and 52.57%/53.20% in $F_1$ on the development/test set, suggesting that this is a challenging task even with ground-truth erroneous spans.

## 5.5 Error Correction

**Task Definition.** We define this task as a sequence-to-sequence generation task. The input to the encoder is an erroneous text where erroneous spans are masked while the output is the corrected text.

**Models.** We use BART as the backbone model for this task. Each erroneous span is replaced with [MASK]. Additionally, we also use an end-to-end BERT-GEC model [15], which directly transforms an erroneous text into a correct sequence without erroneous spans.

**Results.** We do not use conventional text generation evaluation metrics, e.g., BLEU [25], ROUGE [17]. Instead, we focus on the correction to the words in erroneous spans. Particularly, we only consider words in erroneous spans and use them to calculate precision (the number of words correctly corrected vs. all corrections in erroneous spans) and recall (the number of words correctly corrected

vs. the number of words in erroneous spans). Results in Table 2 show that BERT-GEC achieves a very low $F_{0.5}$ of 1.01% and 1.33%, while the erroneous-span-masked BART gains 15.12% and 16.01% on the development set and test set respectively.

## 5.6 Generation Pathology Mitigation

**Task Definition.** Different from GEC, original instances in our dataset are generated from generative PLMs. We hence want to improve the generation capability of generative PLMs so that they could be able to avoid making errors like those annotated in the training data. TGEA 2.0 contains annotations, corrections, rationales of the errors exhibited in language models, providing an opportunity for language models to learn from their errors and corresponding human corrections, in addition to learning from correct texts via self-supervised learning. In order to automatically evaluate how much generative PLMs can learn from TGEA 2.0, we propose two subtasks: pairwise comparison and word prediction. We leave human evaluation on this task to our future work.

- **Pairwise Comparison.** We provide a validation and test set, both of which consist of contrastive pairs $(t_1, t_2)$ where $t_1/t_2$ are either a machine-generated erroneous text or a human-corrected text of the erroneous text. This subtask is partially inspired by previous contrastive evaluations [31, 41, 11]. With this subtask, we want to test whether PLMs enhanced with error-annotated data are able to distinguish pathological outputs from correct outputs.

- **Word Prediction.** Similar to pairwise comparison, we provide a development/test set containing sentences where specific words are masked. We randomly sample 10,000 sentences from Chinese news, wikipedia and web fictions. We then use three different Chinese word segmenters: PKUSEG [20], Jieba[2] and THULAC [32] to segment sentences. 3,024/3,025 sentences are kept for the development/test set, where the last word to be predicted of each sentence can be consistently identified by the three word segmenters (inspired by Chinese WPLC [10]).

**Models.** We propose two methods for these two subtasks using GPT-2 [27] as the baseline: generative fine-tuning and mixed fine-tuning. In the first method, we fine-tune the decoder-only generative baseline on correct sentences (127,684 correct instances judged by annotators) as a causal language model training (additional results are provided in Appendix A.3). For the pairwise comparison subtask, we use the fine-tuned models to estimate the probability of two sentences in each pair. The sentence with a higher probability in a pair is regarded as correct sentence. For the word prediction subtask, the fine-tuned models are used to predict the last word of each sentence from the test set.

The second method, mixed fine-tuning, is inspired by LaMDA [33]. In this method, we fine-tune the baseline in a generative-discriminative-mixed way. In order to enable discriminative fine-tuning, we append an indicator (𝟙 for correct sentence and 𝟘 for erroneous sentence) to each sentence from the contrastive pairs (28,818 in total) in the following manner: <erroneous sentence> 𝟘; <human-corrected sentence> 𝟙. During the generative fine-tuning, all sentences in all contrastive pairs transformed in the aforementioned way are fed into the baseline for causal language model training. The generative loss is the sum of per-token cross-entropy loss over all tokens while the discriminative loss is the sum of per-indicator prediction loss (i.e., $-\log p(\text{indicator}|\text{context})$ where context is the corresponding erroneous/human-corrected sentence) over all instances. The mixed fine-tuning optimizes the generative and discriminative loss together.

After the mixed fine-tuning, for the pairwise comparison subtask, we use the fine-tuned model to predict the label (𝟘 or 𝟙) for each input sentence. For the word prediction task, we first use the fine-tuned model to predict candidate words at the last word position with beam search. We then place each candidate word in the last word position and append the indicator "𝟙" to that word. The candidate word with the highest sentence-level probability is the final predicted word.

**Results.** Results on these two subtasks are shown in Table 3. Generative fine-tuning significantly improves the performance on word prediction but degrades the performance on pairwise comparison. Mixed fine-tuning achieves an improvement of 18.18 points over the GPT-2 baseline on pairwise comparison and 1.55 points on word prediction, indicating discriminative fine-tuning benefits pairwise comparison. Fine-tuning the baseline on correct sentences before the mixed fine-tuning achieves an

---

[2]https://github.com/fxsjy/jieba

| | Pairwise Comparision | | | | Word Prediction | | | | | |
| | Dev | | Test | | Dev | | | Test | | |
| Model | Acc | +δ (↑) | Acc | +δ (↑) | Top-1 | Top-3 | +δ (↑) | Top-1 | Top-3 | +δ (↑) |
|---|---|---|---|---|---|---|---|---|---|---|
| GPT-2 | 55.85 | 0 | 55.20 | 0 | 32.41 | 39.42 | 0 | 30.73 | 37.19 | 0 |
| GF-GPT-2-C | 50.44 | -5.44 | 50.32 | -4.88 | **41.18** | **52.42** | **8.77** | **41.23** | **52.32** | **10.50** |
| MF-GPT-2 | **73.17** | **17.32** | 73.38 | 18.18 | 33.86 | 44.00 | 1.45 | 32.28 | 42.75 | 1.55 |
| -w BC | 72.96 | 17.11 | **74.13** | **18.93** | 34.36 | 46.15 | 1.95 | 34.35 | 45.71 | 3.62 |
| Human | 87.32 | - | 86.79 | - | - | - | - | - | - | - |

Table 3: Results on the pairwise comparison and word prediction tasks. Prefix GF/MF- denote the generative/mixed fine-tuning. -C: fine-tuning on correct sentences. -w BC: fine-tuning the baseline on correct sentences before the mixed fine-tuning.

additional improvement of 0.75 points over the mixed fine-tuning on pairwise comparison and 2.07 points on word prediction.

### 5.7 Discussion on Data Unbalance

Diagnosis tasks, such as erroneous text detection, MiSEW extraction, suffer a data unbalance issue in class distribution even the alpha-balanced loss is used. The reasons for this, we conjecture, could be twofold. First, the erroneous sentences chosen from those generated by causal language models are also challenging for masked language models. Second, the difference between an erroneous and human-corrected text are usually a few characters while the average number of characters in erroneous sentences is around 44. These differences (errors) can only be detected and corrected with long-distance dependencies. Data augmentation methods could be used on TGEA 2.0 to balance the ratio of correct and erroneous sentences. Specifically, we can synthesize additional erroneous sentences based on MiSEWs or use a trained sequence-to-sequence model to convert generated texts into erroneous texts. We leave this to our future work.

### 5.8 Analysis on Generation Pathology Mitigation

We fine-tuned two larger PLMs, PanGu-$\alpha$ (2.6B) and CPM (2.6B), with mixed-tuning on TGEA 2.0 to see if fine-tuning on the dataset helps them to avoid making errors. First, we randomly selected 100 prompts that cause models to generate erroneous sentences for both PanGu-$\alpha$ (2.6B) and CPM (2.6B). We then use the best decoding strategies as mentioned in Section 3.1 for the two models to generate sentences and asked reviewers to evaluate them. We find that the percentages of erroneous sentences of PanGu-$\alpha$/CPM drop from 22.1%/17.62% to 14%/13% after being fine-tuned on our dataset.

## 6 Conclusions and Future Work

In this paper, we have presented TGEA 2.0, the largest Chinese dataset with fine-grained manual annotations on machine texts generated by PLMs across different domains, types of prompts, models and decoding strategies. For each PLM-authored erroneous text, well-trained annotators detect erroneous spans, extract MiSEWs, categorize errors and correct them, under a quality control protocol with feedback loop. Using TGEA 2.0 as a benchmark testbed, we define and evaluate 5 diagnosis tasks and 2 pathology mitigation tasks. Low results of these tasks suggest that TGEA 2.0 is a challenging dataset which could promote automatic diagnosis and pathology mitigation study for text generation from pretrained language models. The current version of the dataset covers 4 Chinese PLMs. We plan to extend the dataset to cover larger PLMs (>100B parameters) recently released. We would also like to organize shared tasks based on the dataset to promote automatic diagnosis and pathology mitigation study for text generation from pretrained language models.

## Acknowledgements

The present research was supported by Huawei. We would like to thank the anonymous reviewers for their insightful comments.

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
