# TGEA 2.0 Supplementary Materials

## A   Appendix

### A.1   Comparison Results in Data Collection & Quality Control and Error/MiSEW Distributions

Here we provide empirical results in the initial study, which support our strategies in data collection (Table 1 - 2) and quality control (Table 3). We also visualize the error distributions of the three datasets in Figure 1. The distribution of MiSEW over the number of tokens contained in each MiSEW is shown in Figure 2.

|            | Nominal | Phrasal | Sentential |
|------------|---------|---------|------------|
| NEZHA-Gen  | 14      | 19      | 15         |
| GPT-2      | 19      | 14      | 13         |
| PanGu-$\alpha$ | 14  | 19      | 15         |
| CPM        | 12      | 14      | 13         |
| total      | 59      | 66      | 56         |

Table 1: The number of erroneous texts generated by different PLMs with different types of prompts (40 prompts for each prompt type).

| Strategies | Nezha-Gen | | | | GPT-2 | | | | CPM | | | | PanGu-$\alpha$ | | | |
|------------|---|----|---|----|----|----|----|----|----|----|----|----|---|----|----|----|
|            | N | P  | S | T  | N  | P  | S  | T  | N  | P  | S  | T  | N | P  | S  | T  |
| $p$=0.9 $t$=0.9 | 14 | 23 | 7 | 44 | 10 | 21 | 7 | 38 | 15 | 29 | 10 | 54 | 8 | 24 | 8 | 40 |
| $p$=0.9 $t$=0.8 | **7** | **13** | **5** | **25** | 11 | 19 | 8 | 38 | 13 | 27 | 9 | 49 | 5 | 25 | 9 | 39 |
| $p$=0.8 $t$=0.9 | 9 | 12 | 5 | 26 | 12 | 17 | 5 | 34 | 13 | 24 | 8 | 45 | 5 | 21 | 6 | 32 |
| $p$=0.8 $t$=0.8 | 9 | 14 | 6 | 29 | **8** | **15** | **6** | **29** | **13** | **20** | **7** | **40** | **6** | **18** | **6** | **30** |
| $k$=30 $t$=0.9 | 8 | 17 | 9 | 34 | 14 | 23 | 10 | 47 | 13 | 26 | 10 | 49 | 9 | 22 | 10 | 41 |

Table 2: The number of erroneous texts generated with different decoding strategies. N: nominal prompts. P: phrasal prompts. S: sentential prompts. T: total. $t$: sampling temperature. $p$: Top-$p$ sampling. $k$: Top-$k$ sampling.

|                                   | Metrics     | 50 texts | 150 texts | 800 texts |
|-----------------------------------|-------------|----------|-----------|-----------|
| Erroneous text detection          | Accuracy    | 66.9     | 72.1      | 76.0      |
|                                   | $F_1$       | 65.2     | 72.3      | 74.1      |
| Erroneous span location           | $F_1$       | 60.0     | 61.0      | 89.8      |
| Level-1 error type classification | Accuracy    | 65.9     | 84.1      | 88.4      |
| Level-2 error type classification | Accuracy    | 50.1     | 72.3      | 72.1      |
| MiSEW extraction                  | Fuzzy match | 81.7     | 70.8      | 80.3      |
|                                   | $F_1$       | 76.3     | 70.4      | 77.3      |
|                                   | Avg         | 58.9     | 71.9      | 79.7      |

Table 3: Performance (%) of annotators in the pre-annotation stage.

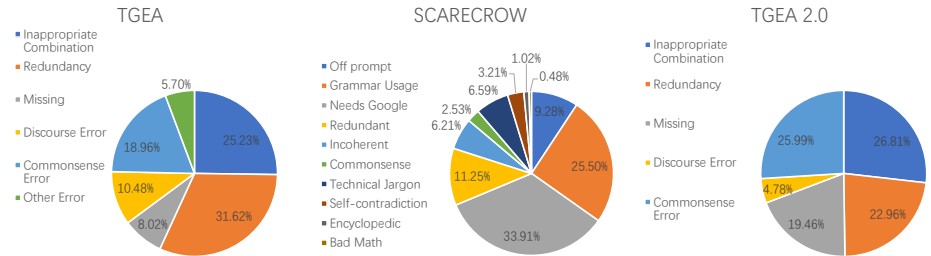

Figure 1: The error distribution over the 10 error types of SCARECROW vs. those over the level-1 error types of TGEA and TGEA 2.0.

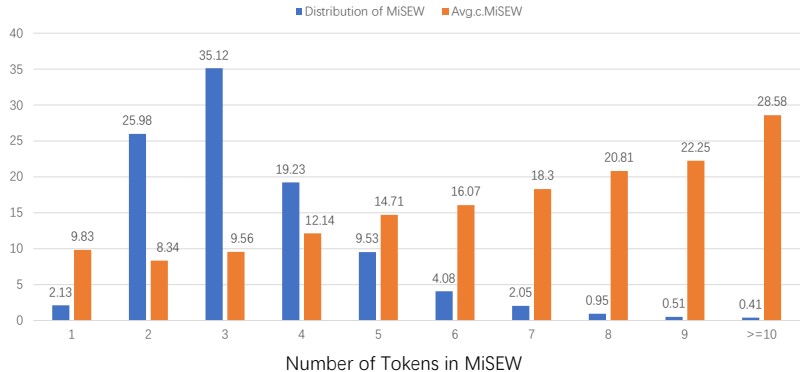

Figure 2: The distribution of MiSEW over the number of tokens contained in each MiSEW. Avg.c.MiSEW: the average number of characters in MiSEW.

## A.2 Experiment Settings

We have fine-tuned several commonly used Chinese PLMs as baselines. RobERTa[1] and MacBERT[1] [1] have a similar model structure of Transformer encoder with 12 layers and attention heads; GPT-2[2] [2] uses a 12-layer Transformer decoder; BART [5] is built on 6-layer encoder-decoder Transformer. All models have 12 attention heads and the hidden size is 768. Models are implemented based on HuggingFace transformers [7]. Detailed statistics for the proposed tasks are shown in Table 4.

| Tasks | Trrain | Dev | Test |
|---|---|---|---|
| Erroneous Text Detection | 156,502 | 19,563 | 19,564 |
| MiSEW Extraction | 28,818 | 3,602 | 3,601 |
| Erroneous Span Location | 33,666 | 4,181 | 4,220 |
| Error Type Classification | 33,666 | 4,181 | 4,220 |
| Error Correction | 28,818 | 3,602 | 3,601 |
| Pairwise Comparision | 156,502 | 3,602 | 3,601 |
| Word Prediction | 156,502 | 3,024 | 3,025 |

Table 4: Statistics for each task.

We train these models on 8 Tesla P100 with 16G memory. The hyperparameters and training time per epoch for each task are shown in Table 5.

## A.3 Additional Benchmark Results

**Diagnosis Tasks.** We have reported the results of baseline models in Section 5. Here, we show more results of Erroneous Sentence Location, MiSEW Extraction, Erroneous Span Detection and Error

---

[1]https://huggingface.co/hfl/chinese-roberta-wwm-ext

[2]https://huggingface.co/uer/gpt2-chinese-cluecorpussmall

| | epoch | learning rate | batch size | time/epoch (minutes) |
|---|---|---|---|---|
| Erroneous Sentence Detection | 3 | $2 \times 10^{-5}$ | 48 | 6.83 |
| MiSEW Extraction | 3 | $2 \times 10^{-5}$ | 32 | 1.33 |
| Erroneous Span Detection | 5 | $2 \times 10^{-5}$ | 32 | 1.86 |
| Error Type Classificaion | 4 | $2 \times 10^{-5}$ | 48 | 1.54 |
| Error Correction | 3 | $5 \times 10^{-5}$ | 12 | 3.48 |
| Pathology Mitigation | 3 | $5 \times 10^{-5}$ | 8 | 3.79 |

Table 5: Hyperparameters and training time for each task.

Type Classification in Table 6. First, we detect erroneous sentences using a combination of focal loss and $\alpha$-balanced loss, and find $\alpha$-balanced loss alone outperforms other methods. Then we treat erroneous span detection and MiSEW extraction as a joint sequence labeling task and two separate tasks. The results in Table 6 show that these two methods are not better than the pipleline method reported in Table 2 in Section 5, where ground-truth MiSEWs benefit the location of erroneous spans. Finally, for error type classification, we use three different methods: (1) adding an error indicator embedding to words in erroneous spans; (2) replacing segment ids with erroneous spans; and (3) a combination of method (2) and (3). We find the method (3) achieves the best performance.

**Pathology Mitigation Tasks.** We have also fine-tuned the decoder-only generative baseline GPT-2 in three ways: (1) fine-tuning on correct sentences (127,684 instances), (2) fine-tuning on both correct and erroneous sentences (156,502 sentences) and (3) fine-tuning on both correct and human-corrected sentences (156,502 correct sentences). All fine-tunings are performed as a causal language model training. The results are shown in Table 7, from which we find that among the three ways of generative fine-tuning, fine-tuning on correct sentences is marginally better than the other two ways.

| Task | Model | Dev | | | | Test | | | |
|---|---|---|---|---|---|---|---|---|---|
| | | A (%) | P (%) | R (%) | $F_1$ (%) | A (%) | P (%) | R (%) | $F_1$ (%) |
| | RoBERTa | 81.84 | 55.17 | 7.41 | 13.07 | 81.99 | 58.35 | 7.66 | 13.54 |
| | -focal loss -$\alpha$=0.5 | 81.87 | 56.72 | 6.26 | 11.57 | 81.93 | 58.13 | 6.74 | 12.09 |
| | -focal loss -$\alpha$=0.3 | 80.52 | 44.19 | 22.07 | 29.44 | 80.47 | 43.8 | 21.26 | 28.62 |
| | -$\alpha$=0.3 | 80.48 | 44.09 | 22.35 | 29.66 | 80.50 | 44.08 | 21.93 | 29.29 |
| Erroneous Text Detection | MacBERT | 81.92 | 57.14 | 7.22 | 12.82 | 82.07 | 59.96 | 7.94 | 14.02 |
| | -focal loss -$\alpha$=0.5 | 81.90 | 58.05 | 6.11 | 11.05 | 82.05 | 61.64 | 6.69 | 12.07 |
| | -focal loss -$\alpha$=0.3 | 80.28 | 43.44 | 23.54 | 30.54 | 80.37 | 43.68 | 22.84 | 30.00 |
| | -$\alpha$=0.3 | 80.07 | 42.8 | 24.43 | 31.11 | 80.35 | 43.83 | 23.84 | 30.88 |
| MiSEW Extraction (joint) | RoBERTa | 72.37 | 42.59 | 23.15 | 26.61 | 72.87 | 42.55 | 22.74 | 26.28 |
| | MacBERT | 72.34 | 40.92 | 22.63 | 25.76 | 72.75 | 40.20 | 22.01 | 25.21 |
| Erroneous Span Location (joint) | RoBERTa | 75.45 | 2.03 | 1.59 | 1.60 | 75.48 | 2.15 | 1.46 | 1.55 |
| | MacBERT | 75.59 | 3.05 | 2.04 | 2.23 | 75.61 | 2.71 | 1.91 | 2.03 |
| | RoBERTa | 75.36 | 0.00 | 0.00 | 0.00 | 75.40 | 0.00 | 0.00 | 0.00 |
| Erroneous Span Location (separate) | RoBERTa-$\alpha$=0.35 | 75.41 | 1.02 | 0.57 | 0.65 | 75.47 | 1.24 | 0.74 | 0.85 |
| | MacBERT | 75.36 | 0.00 | 0.00 | 0.00 | 75.40 | 0.00 | 0.00 | 0.00 |
| | MacBERT-$\alpha$=0.35 | 75.46 | 1.27 | 0.83 | 0.89 | 75.45 | 1.09 | 0.72 | 0.78 |
| | RoBERTa-add | 33.70 | 31.89 | 27.02 | 25.08 | 32.68 | 34.14 | 27.11 | 25.19 |
| | RoBERTa-replace | 41.45 | 43.30 | 35.63 | 36.13 | 40.57 | 43.99 | 35.35 | 36.11 |
| Error Type Classification | RoBERTa-add-replace | 52.43 | **53.89** | 51.97 | **52.57** | 52.70 | **55.35** | 52.21 | **53.20** |
| | MacBERT-add | 32.98 | 31.94 | 26.87 | 25.48 | 32.77 | 31.51 | 27.35 | 25.69 |
| | MacBERT-replace | 46.67 | 49.22 | 42.84 | 44.59 | 46.67 | 50.04 | 43.15 | 44.95 |
| | MacBERT-add-replace | 52.12 | 52.74 | **52.51** | 52.56 | **53.11** | 53.39 | **53.05** | 53.11 |

Table 6: Additional benchmark results on diagnosis tasks. -add: adding an error indicator embedding. -replace: replacing segment ids with error indicator ids. A: Accuracy. P: Precision. R: Recall.

| | Pairwise Comparision | | | | Word Prediction | | | | | |
|---|---|---|---|---|---|---|---|---|---|---|
| | Dev | | Test | | Dev | | | Test | | |
| Model | Acc | $+\delta$ ($\uparrow$) | Acc | $+\delta$ ($\uparrow$) | Top-1 | Top-3 | $+\delta$ ($\uparrow$) | Top-1 | Top-3 | $+\delta$ ($\uparrow$) |
| GPT-2 | 55.85 | 0 | 55.20 | 0 | 32.41 | 39.42 | 0 | 30.73 | 37.19 | 0 |
| GF-GPT-2-CE | 50.24 | -5.61 | 49.65 | -5.55 | 40.86 | 51.43 | 8.45 | 40.91 | 52.23 | 10.18 |
| GF-GPT-2-CC | 50.58 | -5.27 | 50.57 | -4.63 | 40.81 | 51.22 | 8.40 | 40.80 | 51.20 | 10.07 |
| GF-GPT-2-C | 50.44 | -5.41 | 50.32 | -4.88 | **41.18** | **52.42** | **8.77** | **41.23** | **52.32** | **10.50** |

Table 7: Additional benchmark results on pathology mitigation tasks. Prefix GF- denotes generative fine-tuning. -C: fine-tuning on correct sentences. -CE: fine-tuning on both correct and erroneous sentences. -CC: fine-tuning on both correct and human-corrected sentences.

## A.4  Dataset Examples

We show examples in Table 8. More samples are provided at the project website: https://github.com/tjunlp-lab/TGEA/.

| Level-1 Error Type | Level-2 Error Type | Example |
|---|---|---|
| Inappropriate Combination | Subject-Predicate | 大洪水就是这样压迫[摧毁]我们的,是在大雨的暴力之下夺去了你的性命。
That's how the Flood oppress [destroy] us, it will take your life under the violence of the heavy rain.
MiSEW: {大洪水, 压迫, 我们} /
{the Flood, oppress, us} |
| | Predicate-Object | 不等他反应过来,徐河就已经双脚猛蹬在墙壁上,浑身挣脱开了无数鞭痕[绳子],一屁股摔倒在地。
Before he reacted, Xu He was already slamming his feet on the wall, wrench away from countless whiplash marks [ropes] and falling to the ground.
MiSEW: {徐河, 浑身挣脱开, 鞭痕} /
{Xu He, wrench away from, whiplash marks} |
| | Subject-Object | 葡萄皮一般是成熟的苹果皮 [葡萄皮], 采用现代工艺制作而成。
Grape skins are generally ripe apple skins [grape skins], which are made by modern techniques.
MiSEW: {葡萄皮, 是, 苹果皮} /
{Grape skins, are, apple skins} |
| | Modifier | 因为氯化钾可以杀死大量有害昆虫、微生物和叶斑病病原体, 而且能杀灭对人体有益的[有害的]杂草。
Because potassium chloride can kill a large number of harmful insects, microorganisms and leaf spot pathogens, it can kill weeds that are beneficial [harmful] to the human body.
MiSEW: {氯化钾, 杀灭, 对人体, 有益的, 杂草} /
{potassium chloride, kill, to the human body, beneficial, weeds } |
| | Function Word | 我们面对大数据行情的时候, 很多人都想要入门, 可是在[应]选择什么样的机器学习框架呢?
When we face the market of big data, many people want to get started, but what kind of machine learning framework are [should] we choose?
MiSEW: {我们, 在, 选择, 什么样的, 学习框架} /
{we, are, choose, what kind of, machine learning framework} |
| Missing | Subject | 与治疗脱发药物非那甾胺和相似的是, [这药]抑制毛囊细胞活性以达到控油、减少掉头发的目的。
Similar to the hair loss medication finasteride, [this medication] inhibits hair follicle cell activity to achieve the goals of oil control and reduce hair loss.
MiSEW: {抑制, 活性} /
{inhibits, activity} |
| | Predicate | 我当初用着便宜的显示器[导致]眼睛疼的不行。
I use a cheap monitor [which make] my eyes hurt.
MiSEW: {显示器, 眼睛, 疼} /
{monitor, my eyes, hurt } |
| | Object | 一、嵩山派的五绝, 包括鸠摩智、段誉和虚竹三人, 分别是金庸小说中最强大的三个武功门派[代表]。
1. The five masters of the Songshan Sect, including Hatsuma Zhi, Duan Yu, and Xuan Zhu, are the three most powerful [representatives] of the martial arts sect in Jin Yong's novels, respectively.
MiSEW: {三人, 分别, 是, 武功门派} /
{three, respectively, are, martial arts sect} |
| | Modifier | 不知道你看过《阿凡达》没有, 里面讲述了一个奇妙的地球[之外]环境, 如果有兴趣可以去看看。
I don't know if you've watched the movie "Avatar", which tells a wonderful environment [outside] the earth, if you are interested, you can go to watch it.
MiSEW: { 《阿凡达》, 地球环境} /
{"Avatar", environment of the earth} |
| | Function Word | 他不知道该怎么回答, 也不敢想太多。"[虽然]我现在不行, 但以后我会努力的!"
He don't know how to answer and don't dare to think too much. "[Although] I can't do it now, I will try it in the future!"
MiSEW: {我, 现在, 但, 我会} /
{i, now, i will} |
| Redundancy | Subject | 鼠[]眼前是一个光滑的石桥, 地面上满是黄色的尘土和淤泥。
Rat [] in front of eyes is a smooth stone bridge, and the ground is full of yellow dust and silt.
MiSEW: {鼠, 眼前, 是, 一个, 光滑的, 石桥} /
{Rat, in front of eyes, is a, smooth, stone bridge} |
| | Predicate | 随后在小惠的不断劝说下, 会议最终定为1号提案;但是随后李定宣布[]说由于自己还有其他事要做, 因此2号继续开会。
Subsequently, under the continuous persuasion of Xiao Hui, the meeting was finally set as Proposal No.1; But then Li Ding announced [] said that because he had other things to do, he continued to meet on the 2nd.
MiSEW: {宣布, 说} /
{announced, said} |
| | Object | 可是现在他们不在了;就好像这个家里从来不缺少欢笑声和笑声[]一样。
But now they are gone; It was as if there was never a shortage of laughter and laughter [] in this home.
MiSEW: {欢笑声, 笑声} /
{laughter, laughter} |
| | Modifier | 直到殿中只剩下安氏和安荣两人,她们各自诉说着对方的[]哀怨与怨恨。
Until only two people, An Shi and An Rong, were left in the temple, and they told theirs other's [] grievances and resentments.
MiSEW: {她们, 各自诉说, 对方的, 哀怨与怨恨} /
{they, told, theirs, other's, grievances and resentments} |
| | Function Word | 雨水退去,而是[]从地平线上的一个小点开始慢慢消散开了。
The rain receded, but [] slowly disappeared from a small point on the horizon.
MiSEW: {雨水退去, 而是, 从地平线上, 消散} /
{The rain receded, but, disappeared, from the horizon} |

| Level-1 Error Type | Level-2 Error Type | Example |
|---|---|---|
| Discourse Error | Coreference | 我跟女友认识三年多，订婚前夕她妈妈[女友]告诉我说她爸妈逼迫她和我分手，理由嫌弃我没有钱。
I have known my girlfriend for more than three years, and on the eve of our engagement, my girlfriend's mother [girlfriend] told me that her parents force her to break up with me with reasons that I don't have any money.
MiSEW: {她妈妈, 说, 她爸妈, 逼迫, 分手} /
{girlfriend's mother, told, her parents, force, break up} |
| Commonsense Error | Space | 据介绍，安居镇[它]是安徽省淮北地区最大的开发建设县之一，全镇现有22万人口，其中绝大多数是外出务工经商者。
According to reports, Anju [this] town is one of the largest developing counties in huaibei district of Anhui Province. The town has a population of 220,000, the vast majority of whom are migrant workers and businessmen.
MiSEW: {安居镇, 是, 安徽省, 建设县} /
{Anju town, is, Anhui Province, developing counties } |
| | Time | 起源时间:20[19]世纪初期(1840年前后).
Origin Time: Early 20th [19th] century (around in 1840).
MiSEW: {20世纪, 1840年} /
{20th, in 1840} |
| | Number | 五寨乡共辖9个行政村，其中有5个自然村、2[4]个人工村。
Wuzhai Township has a total of 9 administrative villages, including 5 natural villages and 2 [4] artificial villages.
MiSEW: {9个, 5个, 2个} /
{9, 5, 2} |
| | Motivation | 近日，沈海磊告诉记者，有两个方面的原因：1、从他开始在北京发展以来，北京就是他一直想要去追求和占据[居住]的地区。
Recently, Shen Hailei told reporters that there are two reasons: 1. Since he started developing in Beijing, Beijing is the area he has always wanted to pursue and occupy [live in].
MiSEW: {想要, 追求, 占据} /
{want, pursue, occupy} |
| | Emotional Reactions | 我们可以看到他从小就不得志，但是却对父亲一直很敬重，甚至有点恨铁不成钢[过分敬重]。
We can see that he has been demoralized since he was a child, but he has always had great respect for his father, even a little "wish iron could turn into steel at once" [too much respect].
MiSEW: {对, 父亲, 恨铁不成钢} /
{for, his father, "wish iron could turn into steel at once"} |
| | Causation | 毛类禽流感疫情以每年10% 15%的速度增长，一般呈下降[上升]趋势。
Outbreaks of gross avian influenza are growing at a rate of 10% to 15% per year and generally show a descend [upward] trend.
MiSEW: {10% 15%增长, 下降} /
{10% to 15%, growing, descend} |
| | Taxonomy | 没有比玫瑰木更好的衣料了！
There is no better lining [wood] than rosewood!
MiSEW: {没有比 玫瑰木 衣料} /
{no, rosewood, lining} |
| | Behaviors | 据介绍，广州市眼科专家在给患者做检查时发现，当前部分青少年对酒精中毒[预防近视]有一定认识，但并没有注意到这些问题。
According to reports, Guangzhou ophthalmologists find that some adolescents currently have a certain understanding of the alcoholism [myopia prevention] when examining patients, but they have not noticed these problems.
MiSEW: {眼科专家, 发现, 青少年, 对, 酒精中毒, 有, 一定认识} /
{ophthalmologists, find, adolescents, have, a certain understanding, of, the alcoholism} |

Table 8: Examples of TGEA 2.0. Red words are erroneous words. Words in "[]" are corrections to erroneous words (Empty "[]" denoting deletion).

We show examples of model predictions for each benchmark task in Table 9.

| | 后来,他又收养了三男三女四个孩子。
Later, he adopted a total of four children,
three boys and three girls. | |
|---|---|---|
| **Task** | **Model Prediction** | **Human Annotation** |
| Erroneous Text Detection | Incorrect | Incorrect |
| MiSEW Extraction | {收养了, 三男, 三女, 四个孩子} /
{adopted, three boys, three girs,
a total of four childen} | {三男, 三女, 四个孩子} /
{three boys, three girs,
a total of four childen} |
| Erroneous Span Location | 孩子 / childen | 四 / four |
| Error Type Classification | Commonsense
Error | Commonsense
Error |
| Error Correction | 后来,他又收养了
三男三女三个孩子。
Later, he adopted
a total of three children,
three boys and three girls. | 后来,他又收养了
三男三女六个孩子。
Later, he adopted
a total of six children,
three boys and three girls. |
| | 如果你觉得她对你不重要，而且又感受不到，那就可以分手了。
If you feel she is not important to you and you don't feel,
it's time to break up. | |
| **Task** | **Model Prediction** | **Human Annotation** |
| Erroneous Text Detection | Incorrect | Incorrect |
| MiSEW Extraction | {觉得, 她, 对你, 不重要, 感受, 不到} /
{feel, she, is, not important, to you,
don't, feel} | {感受, 不到} /
{don't, feel} |
| Erroneous Span Location | 到 / feel | 到 / feel |
| Error Type Classification | Missing | Missing |
| Error Correction | 如果你觉得她对你不重要，而且又
感受不到她，那就可以分手了。
If you feel she is not important
to you and you don't feel her,
it's time to break up. | 如果你觉得她对你不重要，而且又
感受不到爱，那就可以分手了。
If you feel she is not important
to you and you don't feel love,
it's time to break up. |
| | 林尾镇林尾镇,是下辖的一个乡镇级行政单位。
Linwei Town Linwei Town is a township
level administrative unit under its jurisdiction. | |
| **Task** | **Model Prediction** | **Human Annotation** |
| Erroneous Text Detection | Incorrect | Incorrect |
| MiSEW Extraction | {林尾镇, 是, 行政单位} /
{Linwei Town, is, administrative unit} | {林尾镇, 林尾镇, 是, 单位} /
{Linwei Town, Linwei Town, is, unit} |
| Erroneous Span Location | 林尾镇 / Linwei Town | 林尾镇 / Linwei Town |
| Error Type Classification | Reduancy | Reduancy |
| Error Correction | 林林尾镇,是下辖的一个乡镇级行政单位。
LinLinwei Town is a township level
administrative unit under its jurisdiction. | 林尾镇,是下辖的一个乡镇级行政单位。
Linwei Town is a township level
administrative unit under its jurisdiction. |

Table 9: Examples of model predictions.

## A.5   Sources of the Training Data of the Used PLMs

We've manually checked TGEA 2.0 and found that more than 99% texts are simplified Chinese. However, some machine-authored texts are traditional Chinese, as shown in Table 10. We conjecture that sources of the training data of the used 4 PLMs contain traditional Chinese texts. We hence provide simple data cards for the 4 PLMs in Table 11.

| Model | Traditional Chinese Examples in TGEA 2.0 |
|---|---|
| NEZHA-Gen | 所以,了保持流動性不會過剩,必要把流性進行適當調節。
Therefore, in order to keep the liquidity from being excessive, the liquidity must be properly adjusted. |
| GPT-2 | 這些人大部份都從小就被送去國外接受高等教育,所以學到了非常先進的知和思想,加上身一群有見解、有智慧、會播知的人,自然會成為佼佼者。
Most of these people have been sent abroad for higher education when they were children, so they have learned very advanced knowledge and ideas, and they will naturally become outstanding as a group of people with insight, wisdom and knowledge. |
| CPM | 很多人也不知道怎麼去想這問題,為什麼就會有這樣的結論呢?首先先給自己找個理由吧!
Many people don't know how to think about this issue. Why do they come to such a conclusion? First, find a reason for yourself! |
| PanGu-$\alpha$ | 每個區域都有明確的表格,讓觀眾可以清楚知道自己是否被限制在某個範圍內;而每部劇作也均會根據不同的要求對表演進行調整,因此觀眾能夠自由地觀看和選擇任何一部作品。
Each area has a clear table, so that the audience can clearly know whether they are confined to a certain range; and each play will also adjust the performance according to different requirements, so the audience is free to watch and choose any one works. |

Table 10: Traditional Chinese Examples of TGEA 2.0.

| Model | Model Location | Data Source | Writing System |
|---|---|---|---|
| NEZHA-Gen | https://github.com/ huawei-noah/ Pretrained-Language-Model/ tree/master/ NEZHA-Gen-TensorFlow | Chinese Wikipedia[3]: encyclopedia containing 1,067,552 articles. The cleaned data contain both simplified and traditional Chinese texts with roughly 202M tokens. | Simplified / Traditional Chinese |
| | | Baidu Baike[4]: webpages from the Baidu Baike with more than 15.4 million articles. The cleaned corpus contains 4,734M tokens. | |
| | | Chinese News: multiple news websites (e.g. Sina News). The cleaned corpus contains 5,600M tokens. | |
| GPT-2 | https://github.com/ ghosthamlet/ gpt2-ml-torch | THUCNews [6] and NLP Chinese Corpus [9]. The size of cleaned the corpus is around 15GB. | Simplified / Traditional Chinese |
| CPM | https://github.com/ TsinghuaAI/ CPM-1-Generate | Encyclopedia (40GB), Webpage (39GB), Story (10GB), News (10GB) and Dialog (1GB). | Simplified / Traditional Chinese |
| PanGu-$\alpha$ | https://github.com/ huawei-noah/ Pretrained-Language-Model/ tree/master/PanGu-%CE%B1 | Public datasets (27.9GB): 15 public datasets including DuReader [4], BaiDuQA[5], CAIL2018 [8], Sogou-CA[6], etc. | Simplified / Traditional Chinese |
| | | Encyclopedia (22GB): Baidu Baike, Sogou Baike, etc. | |
| | | e-Books (299GB): e-Books on various topics (e,g., novels, history, poetry, ancient prose, etc.). | |
| | | Common Crawl (714.9GB): Web data Common Crawl (snapshots from January 2018 to December 2020). | |
| | | News (35.5GB): News data from 1992 to 2011. | |

Table 11: Data sources of the used 4 PLMs.

---

[3]https://zh.wikipedia.org/wiki/

[4]https://baike.baidu.com/

[5]http://research.baidu.com/Downloads

[6]http://www.sogou.com/labs/resource/ca.php

## B  Project Statement

**Project Website: https://github.com/tjunlp-lab/TGEA/**

**Dataset.** Prompts are collected from online websites while continuations to prompts are generated by publicly available Chinese PLMs. All the annotations are verified by automatic and manual check & review.

**Accessibility.** The annotated dataset with benchmark models will be publicly available at minspore and Github.

**Licence.** The dataset will be released under the CC BY-SA 4.0 license for general research purposes.

The authors declare that they bear all responsibility for violations of rights related to this dataset.

## C  Datasheet

### C.1  Motivation

**For what purpose was the dataset created? Was there a specific task in mind? Was there a specific gap that needed to be filled? Please provide a description**

We create the dataset to diagnostically analyze and improve the capability of PLMs in text generation. Although generative PLMs are capable of generating texts that are sometimes not distinguishable from human-written texts, they suffer from quality issues (e.g, grammatical correctness, semantic coherence). In comparison to error-annotated datasets built on human-written texts, the current two error-annotated datasets (TGEA and SCARCEROW) on machine-generated texts are small. TGEA 2.0 is curated to fill this dataset size gap to enable automatic diagnosis on machine texts with five diagnosis tasks. The second group of tasks are pathology mitigation tasks which have not been defined in the previous two datasets.

**Who created this dataset (e.g., which team, research group) and on behalf of which entity (e.g., company, institution, organization)?**

The dataset has been collectively curated by the Natural Language Processing Lab of Tianjin University and Huawei Noah's Ark Lab.

**Who funded the creation of the dataset? If there is an associated grant, please provide the name of the grantor and the grant name and number.**

The dataset is sponsored by Huawei (No. TC20210528011).

### C.2  Composition

**What do the instances that comprise the dataset represent (e.g., documents, photos, people, countries)? Are there multiple types of instances (e.g., movies, users, and ratings; people and interactions between them; nodes and edges)? Please provide a description.**

The instances of TEGA 2.0 are texts generated by PLMs triggered by natural prompts. As shown in Table 12, each annotated instance contains an erroneous sentence, its corrected version, confidence score of annotators and a list of manual annotations related to each error including: erroneous span, MiSEW, level-1/2 error type. Extra information such as prompt, model, type and domain are also included.

**How many instances are there in total (of each type, if appropriate)?**

TGEA 2.0 consists of 195,629 annotated sentences, including 36,023 erroneous sentences with 42,067 erroneous spans. Detailed statistics are included in Section 4.1.

**Does the dataset contain all possible instances or is it a sample (not necessarily random) of instances from a larger set? If the dataset is a sample, then what is the larger set? Is the sample representative of the larger set (e.g., geographic coverage)? If so, please describe how this representativeness was validated/verified. If it is not representative of the larger set, please describe why not (e.g., to cover a more diverse range of instances, because instances were withheld or unavailable).**

| Annotation Item | Description |
|---|---|
| ID | Sentence ID |
| Context | Erroneous sentence |
| Correction | Corrected sentence |
| Confidence | Confidence score for each annotated sentence |
| Annotation | Erroneous span
MiSEW
Level-1 error type
Level-2 error type |
| Extra Information | Prompt: prompt for text generation
Model: PLM used to generate the paragraph
Type: type of the prompt
Domain: domain of the prompt |

Table 12: Annotation items and description in TGEA 2.0.

The dataset contain instances generated by PLMs triggered by natural prompts. In order to diversify generated texts, multiple types of natural prompts (i.e., nouns, phrases, sentences) are extracted from 6M sentences in 3 domains. Additionally, 4 different PLMs with tailored decoding strategies are explored to generate texts that are representative of texts generated by PLMs under the best setting. Annotated errors cover 5 level-1 error types and 24 level-2 error types.

**What data does each instance consist of? "Raw" data (e.g., unprocessed text or images)or features? In either case, please provide a description.**

Machine-generated texts with manual semantic annotations. Please refer to Table 12 for more details.

**Is there a label or target associated with each instance? If so, please provide a description.**

We annotate each erroneous sentence with rich semantic information, e.g., the corrected sentence, erroneous spans, MiSEWs, level-1/2 error types. Please refer to Table 12 for more details.

**Is any information missing from individual instances? If so, please provide a description, explaining why this information is missing (e.g., because it was unavailable). This does not include intentionally removed information, but might include, e.g., redacted text.**

The information of each instance is self-contained.

**Are relationships between individual instances made explicit (e.g., users' movie ratings, social network links)? If so, please describe how these relationships are made explicit.**

Individual instances are extracted from paragraphs generated by models triggered by different natural prompts.

**Are there recommended data splits (e.g., training, development/validation, testing)?**

We randomly split the dataset into the training/dev/test set according to a proportion of 8:1:1.

**Are there any errors, sources of noise, or redundancies in the dataset? If so, please provide a description.**

The dataset is carefully reviewed and checked automatically and manually with a strict quality control protocol.

**Is the dataset self-contained, or does it link to or otherwise rely on external resources (e.g., websites, tweets, other datasets)?**

The dataset is self-contained.

**Does the dataset contain data that might be considered confidential (e.g., data that is protected by legal privilege or by doctor– patient confidentiality, data that includes the content of individuals' non-public communications)? If so, please provide a description.**

No.

**Does the dataset contain data that, if viewed directly, might be offensive, insulting, threatening, or might otherwise cause anxiety? If so, please describe why.**

No. Machine-generated texts that are offensive are manually filtered out.

**If the dataset does not relate to people, you may skip the remaining questions in this section**

This dataset does not relate to people.

### C.3 Collection

**How was the data associated with each instance acquired? Was the data directly observable (e.g., raw text, movie ratings), reported by subjects (e.g., survey responses), or indirectly inferred/derived from other data (e.g., part-of-speech tags, model-based guesses for age or language)? If data was reported by subjects or indirectly inferred/derived from other data, was the data validated/verified? If so, please describe how.**

The data associated with each instance is manually annotated according to a predefined annotation convention and guideline, which include erroneous spans, corrections, MiSEWs and error types. These items are all directly observable.

**What mechanisms or procedures were used to collect the data (e.g., hardware apparatus or sensor, manual human curation, software program, software API)? How were these mechanisms or procedures validated?**

Our data collection is composed of three stages: (1) natural prompt collection from extracted sentences; (2) generating paragraphs with multiple PLMs under their desirable decoding settings according to given prompts; (3) manual annotation over machine-generated texts. The entire collection procedure is equipped with prompt collection, model selection & setting, annotation convention, error taxonomy and quality control. The annotation convention, prompt selection and error taxonomy are well validated in TGEA [3] while others are extended from TGEA.

**Who was involved in the data collection has process (e.g., students, crowdworkers, contractors) and how were they compensated (e.g., how much were crowdworkers paid)?**

To guarantee annotation quality, we contracted out the data collection to a professional data collection & annotation company SpeechOcean (http://en.speechocean.com/), which is well experienced in data annotation in speech and natural language processing. All annotators of the subcontractor are well trained with our annotation convention in a pre-annotation stage. The cost for each correct instance is annotation 0.3 yuan while 1.8 yuan for each erroneous instance annotation.

**Over what timeframe was the data collected? Does this timeframe match the creation time frame of the data associated with the instances (e.g., recent crawl of old news articles)? If not, please describe the timeframe in which the data associated with the instances was created.**

The data collection process lasted for around 90 days. The natural prompts are extracted from the crawl of recent news, Wikis and web fictions. The selected 4 PLMs are also those recently released (released time ranging from 2019 to 2021).

**Were any ethical review processes conducted (e.g., by an institutional review board)?**

Ethical review was conducted by our subcontractor.

**Was the "raw" data saved in addition to the preprocessed/cleaned/labeled data (e.g., to support unanticipated future uses)? If so, please provide a link or other access point to the "raw" data.**

Yes, the raw data is currently archived and will be released when necessary.

**Is the software used to preprocess/clean/label the instances available? If so, please provide a link or other access point.**

Yes, the annotation software and tools will be available soon.

**Has the dataset been used for any tasks already?**

No, not yet.

**Is there a repository that links to any or all papers or systems that use the dataset?**

No.

**What (other) tasks could the dataset be used for?**

The dataset could be used for other tasks related to machine texts (e.g., machine-generated text detection) or tasks related to the comparison between human-written and machine-generated texts.

**Is there anything about the composition of the dataset or the way it was collected and preprocessed/cleaned/labeled that might impact future uses? For example, is there anything that a dataset consumer might need to know to avoid uses that could result in unfair treatment of individuals or groups (e.g., stereotyping, quality of service issues) or other risks or harms (e.g., legal risks, financial harms)? If so, please provide a description. Is there anything a dataset consumer could do to mitigate these risks or harms?**

No.

**Are there tasks for which the dataset should not be used? If so, please provide a description.**

Tasks that are not related to machine texts.

### C.4    Distribution

**Will the dataset be distributed to third parties outside of the entity (e.g., company, institution, organization) on behalf of which the dataset was created?**

Yes, probably.

**How will the dataset will be distributed (e.g., tarball on website, API, GitHub)? Does the dataset have a digital object identifier (DOI)?**

The dataset will be distributed at https://github.com/tjunlp-lab/TGEA/.

**When will the dataset be distributed?**

The dataset will be released by stages according to the schedule of shared tasks that we plan to organize with the dataset. The full training dataset and a small dev/test dataset are supposed to be released by July, 2022.

**Will the dataset be distributed under a copyright or other intellectual property (IP) license, and/or under applicable terms of use (ToU)? If so, please describe this license and/or ToU, and provide a link or other access point to, or otherwise reproduce, any relevant licensing terms or ToU, as well as any fees associated with these restrictions.**

This dataset is released under the CC BY-SA 4.0 license for general research purposes.

**Have any third parties imposed IP-based or other restrictions on the data associated with the instances? If so, please describe these restrictions, and provide a link or other access point to, or otherwise reproduce, any relevant licensing terms, as well as any fees associated with these restrictions.**

No.

**Do any export controls or other regulatory restrictions apply to the dataset or to individual instances? If so, please describe these restrictions, and provide a link or other access point to, or otherwise reproduce, any supporting documentation.**

No.

### C.5    Maintance

**Who is supporting/hosting/maintaining the dataset?**

The dataset will be hosted on Github and will be maintained by Huibin Ge, Xiaohu Zhao, Chuang Liu, Yulong Zeng, Qun Liu and Deyi Xiong from the Natural Language Processing Lab of Tianjin University and Huawei Noah's Ark Lab.

**How can the owner/curator/manager of the dataset be contacted (e.g., email address)?**

The maintainers can be contacted via email: gehuibin@tju.edu.cn, zhaoxiaohu@tju.edu.cn, liuc_09@tju.edu.cn zengyulong@huawei, qun.liu@huawei and dyxiong@tju.edu.cn.

**Is there an erratum? If so, please provide a link or other access point.**

No.

**Will the dataset be updated (e.g., to correct labeling errors, add new instances, delete instances)? If so, please describe how often, by whom, and how updates will be communicated to users (e.g., mailing list, GitHub)?**

The two teams will continue to update the dataset, including but not limited to scaling the dataset, organizing shared tasks with the dataset, providing new test/dev sets. The updates will be yearly and communicated to users through public shared tasks, GitHub, etc.

**If the dataset relates to people, are there applicable limits on the retention of the data associated with the instances (e.g., were individuals in question told that their data would be retained for a fixed period of time and then deleted)? If so, please describe these limits and explain how they will be enforced.**

N/A.

**Will older versions of the dataset continue to be supported/hosted/maintained? If so, please describe how. If not, please describe how its will be communicated to users.**

Yes, older version is still maintained and updated and will be communicated to users via Github.

**If others want to extend/augment/build on/contribute to the dataset, is there a mechanism for them to do so? If so, please provide a description. Will these contributions be validated/verified? If so, please describe how. If not, why not? Is there a process for communicating/distributing these contributions to other users? If so, please provide a description.**

The current version of the dataset covers 4 Chinese PLMs. We are planning to extend the dataset to cover more PLMs in more different languages recently released. Any potential contributors are welcome to join us to expand the dataset to larger size, to multilingual version, or to jointly organize shared tasks based on the dataset. We also would like to attract more researchers from both academics and industry who are interested in text generation quality of PLMs or large language models to form a consortium and to organize themed events, workshops, etc.