# OpenReview forum: "TGEA 2.0: A Large-Scale Diagnostically Annotated Dataset with Benchmark Tasks for Text Generation of Pretrained Language Models"
_NeurIPS.cc/2022/Track/Datasets_and_Benchmarks — NeurIPS 2022 Datasets and Benchmarks _

### Official Review · Reviewer_FbYQ · 2022-07-24
**The authors present an updated version of a dataset in Chinese that is annotated with errors made in NLG tasks by Pretrained Language Models.**

**Rating:** 7
**Confidence:** 4

**Strengths:**

* Improvement over previous work where authors used the stochastic decoding strategy and repetition penalty which reduced the redundancy related errors frequency and hence also allowed energy to be focused on harder errors.
* The number of annotated samples is large enough to gain confidence and mitigate risk of incorrect conclusions due to spurious correlations.
* Benchmark results are shared on the presented dataset using SoTA models which allows the research community to have solid baselines to compare their research and findings against. Furthermore, by evaluating numerous PLMs on diverse datasets, the work also helps users of PLMs in deciding appropriate model for any given NLG task.


**Weaknesses:**

* The reasoning behind choosing three point scale for asking confidence in annotation instead of standard Likert scale is not provided.
* Any inspiration that was taken from related work is missing for the choice of annotator training methodology.
* Given that beyond a certain threshold, PLMs show emergence capabilities. The work done here is called into question as whether or not the errors patterns in small PLMs (<5B parameters) are also prevalent in Large PLMs (> 100B) like PanGu-α (200B), Wu Dao 2.0 (1.75T).


Grammar errors:
1. L168 "think it"
2. L190 "in three times"


**Additional Feedback:**

Authors, kindly upload your datasets to mindspore as well in due time.

**Clarity:**

The paper is clear and well written. The authors avoid use of complex jargon and present concepts in a simplified manner making it easy for readers to digest the concepts being proposed.

**Correctness:**

Yes, the datasets are constructed in sound manner along with the appropriate experiment design and evaluation methodology.

**Documentation:**

Yes, among the two links shared for dataset access, only Github has the dataset in zip format. The other link does not contain any dataset.
The github has enough documentation and code to be able to reproduce the results. I was able to clone the repository and access the dataset from it. The formatting scripts seems fine, the dataset format looked fine although I could not verify what was inside it as I do not know Chinese.

**Ethics:**

The work complies with Ethics Guidelines of the NeurIPS.

**Relation To Prior Work:**

The work discussed tangents that it is related to as well as how it improves the shortcomings of existing works. There is one aspect whose relation to prior work is overlooked, methodology used to train the annotators that annotated the tasks.

**Summary And Contributions:**

The work builds on by releasing a larger and higher quality version of a previously released dataset named TGEA. It is a comprehensive collection of machine authored texts in Chinese language that have been annotated for errors based on a novel ontology of errors. This ontology is based on data mining for frequently occurring forms of errors followed by supervision by expert annotators. Furthermore, systematic analysis to the annotated errors is performed to reveal patterns that are helpful in gauging the capability of various PLMs on different datasets.

Lastly, the work goes on to validate whether errors found can be fixed automatically with pre-existing large language models and find  it hard to solve by modern means. This paves the way for 1) A dataset to analyze the kind errors PLM makes 2) Developing automated methods that can automatically correct the errors made by PLMs because existing SoTA is not enough to rectify it 3) Benchmark of the performance of SoTA on both diagnostic as well as pathological errors for future works to compare against.

---

> ### Author Response · Authors · 2022-08-18
> **Thanks for comments and suggestions**
>
> Thank you very much for your insightful comments and suggestions. Our answers to your comments and questions are as follows.
>
> Q1: The reasoning behind choosing three point scale for asking confidence in annotation instead of standard Likert scale is not provided.
>
> A1: Annotation Confidence is used in our manual check & review. If an annotator feels unconfident on an annotated sentence, we will ask two reviewers to review the sentence. We choose the three point scale since it is efficient for both annotators and reviewers. But many thanks for your suggestion. We’d like to compare the three-point scale with the standard Likert scale in the next phase of annotation and expansion of our dataset.
>
> Q2: Any inspiration that was taken from related work is missing for the choice of annotator training methodology.
>
> A2: Our annotator training methodology is inspired from the TGEA 1.0 [1], and we’ve made changes to the training methodology to include such as automatic and manual check & review to further guarantee annotation quality. The training methodology is also inspired from our previous dataset annotation (e.g., TED-CDB [2], RiSAWOZ [3]), previous works (e.g., PDTB-3 [4]) and collaboration with Prof. Bonnie Webber who has kindly provided many suggestions and training for the TED-CDB construction. We’ll include prior works that are related to / inspire our annotation methodology in the next version.
>
> Q3: Given that beyond a certain threshold, PLMs show emergence capabilities. The work done here is called into question as whether or not the errors patterns in small PLMs (<5B parameters) are also prevalent in Large PLMs (> 100B) like PanGu-α (200B), Wu Dao 2.0 (1.75T)
>
> A3: Many thanks for this great comment and suggestion. We completely agree with you on this and are planning to extend the dataset to cover larger PLMs (>100B parameters) recently released, to investigate if emergence capabilities also happen in pathological patterns/distributions when language models are scaled to a certain threshold.
>
> Q4: Authors, kindly upload your datasets to mindspore as well in due time.
>
> A4: Due to the procedure and restrictions on data uploading, we are afraid that we could not upload our dataset to mindspore in due time. because of the upload restrictions. The github data repository (https://github.com/tjunlp-lab/TGEA/) is sofar our marjor data and project website, where we have already uploaded the dataset and benchmark codes.
>
> [1] Jie He, Bo Peng, Yi Liao, Qun Liu and Deyi Xiong. TGEA: An Error-Annotated Dataset and Benchmark Tasks for Text Generation from Pretrained Language Models. ACL 2021.
>
> [2] Wanqiu Long, Bonnie Webber, Deyi Xiong. TED-CDB: A Large-Scale Chinese Discourse Relation Dataset on TED Talks. EMNLP 2020.
>
> [3] Jun Quan, Shian Zhang, Qian Cao, Zizhong Li, Deyi Xiong. RiSAWOZ: A Large-Scale Multi-Domain Wizard-of-Oz Dataset with Rich Semantic Annotations for Task-Oriented Dialogue Modeling. EMNLP2020.
>
> [4] Bonnie Webber, Rashmi Prasad, Alan Lee, Aravind Joshi. The Penn Discourse Treebank 3.0 Annotation Manual

---

### Official Review · Reviewer_7aUo · 2022-07-25
**Large-scale dataset for diagnosis and mitigation of Chinese language text generation errors**

**Rating:** 8
**Confidence:** 3

**Strengths:**

The size of the dataset and multiple annotations (erroneous spans and minimal set of error-related words and corrections for these words) provide an excellent basis for studying the nature of grammatical and semantic coherence errors in Chinese language automated text generated by current SOTA models performing at their best. As such, this dataset should be of considerable interest to researchers seeking to understand the persistence of certain error types in machine-generated text, and also, to understand, from objective evidence (vs subjective human evaluations, cf. Clark 2021), what kinds of errors may be indicative of machine-generated outputs and therefore may support the detection of machine-generated text. The further interest of this paper is the attempt to use the error-plus-correction MiSEW pairs to improve the quality of the generated text by reducing errors of these kinds in the output. Another strength of the paper is the clear and comprehensive description of the research process, including the annotation process, which adheres to annotation best practices in many ways (including an indicator of annotator confidence, pre-training to support annotator convergence to a high level of inter-annotator agreement and iterative re-training during annotation, etc).

**Weaknesses:**

It would have been good to see Cohen's Kappa statistics (or similar) cited for the inter-annotator agreement in section 3.3. Average accuracy of annotators is mentioned ("average performance ... increases from 58.9% to 79.7%") and also "inter-annotator disagreement" but not measure of the latter is explicitly provided.

**Additional Feedback:**

In the Conclusion section, I would be interested to hear what the authors see as the next step for their work -- whether it would be creating a further dataset or specific lines of exploration of the TGEA 2.0 dataset they have created.

**Clarity:**

Yes, the paper is very well-written and clear. There are a very few typos/grammatical errors which would be good to correct: line 32, add "on" after "done"; line 39, substitute "with which" for "that"; standardise "rational" to "rationale" throughout.

**Correctness:**

Yes, the claims made in the submission appear correct and the benchmark dataset has been created in an appropriate and correct manner. Substantial care has been given to the impact of variables such as model size, decoding strategy etc on the errors occurring in the generated texts, and analysis of the impact of these variables on the error type frequency and distribution in the dataset.

**Documentation:**

The data collection process is clearly documented. However, it would be useful, from an ethical sourcing perspective, to provide some basic data on the sources of the datasets used to train the 4 PLMs, which are in turn used to generate the sentences curated in this dataset of machine-generated texts. The intended uses of the datasets are clear (machine-generated semantic and grammatical error detection, classification and mitigation) and may additionally support the detection of machine- vs human-generated Chinese language text. A URL is provided for access to the dataset and hosting and licensing details. As far as I can tell, sufficient detail is given to support reproducibility in all aspects.

**Ethics:**

There is a difficult and highly sensitive social and ethical problem that I feel the need to flag, which exists in all current Chinese language corpora used as training data for PLMs and is reflected in this paper. This is the erasure and invisibility of linguistic variability in Chinese language training data through (1) standardisation to the simplified character set where traditional characters occur in the input data, ensuring that traditional characters will not be output by the PLM, and (2) lack of acknowledgement that "Chinese" is linguistically not a single language (as recognised by the variety of ISO codes for Chinese languages in ISO 639-3, including Yue (Cantonese)) and that the written input sources may include text that has variations in word order and grammatical structure typical of, for example, Yue (e.g. Kaltenegger, S. 2020; Yan, J. 2008 (dissertation)). At a minimum, the standardisation of the character set in the underlying data used to train PLMs, and its implications for the output character set, should be explicitly flagged, as well as the geolocations from which the training data was collected. Where appropriate (i.e. if the underlying data is indeed standard Mandarin Chinese -- and the fact that most massive datasets are collected from the web makes it unlikely that the data will include only standard Mandarin Chinese), the language descriptor "Chinese" should be modified by "Mandarin" to acknowledge the existence of other varieties of Chinese.

**Relation To Prior Work:**

Yes, the relationship to prior work is sufficiently discussed in the opening sections of the paper.

**Summary And Contributions:**

This paper contributes to understanding and reducing the text generation errors made by large pre-trained language models. The authors have created the largest Chinese language dataset of machine-authored texts and a substantial subset of the texts (>195k) were manually annotated at a fine-grained level and corrected for text quality issues (grammaticality and semantic coherence). The authors use the annotated data both to benchmark the best performance of 4 major PLMs (with variable architecture and scale) against each top-level error category and to test the extent to which fine-tuning with the human-corrected texts reduces the prevalence of these errors in the PLM outputs.

---

> ### Author Response · Authors · 2022-08-18
> **Thanks for comments and suggestions**
>
> Thank you very much for your insightful comments and suggestions. Our answers to your comments and questions are as follows.
>
> Q1: It would have been good to see Cohen's Kappa statistics (or similar) cited for the inter-annotator agreement in section 3.3. Average accuracy of annotators is mentioned ("average performance ... increases from 58.9% to 79.7%") and also "inter-annotator disagreement" but not measure of the latter is explicitly provided.
>
> A1: The average Cohen's Kappa statistics of 7 reviewers /annotators is  56.6% /49.3%.
>
> Q2: In the Conclusion section, I would be interested to hear what the authors see as the next step for their work -- whether it would be creating a further dataset or specific lines of exploration of the TGEA 2.0 dataset they have created.
>
> A2: The current version of the dataset covers 4 Chinese PLMs. We are planning to extend the dataset to cover larger PLMs (>100B parameters) recently released, to see if any significant changes in error types/distributions happen as language models are scaled. We would also like to organize shared tasks based on the dataset to promote automatic diagnosis and pathology mitigation study for text generation from pretrained language models. Many thanks for this great suggestion. We’ll include our future work plan in the next version.
>
> Q3: There is a difficult and highly sensitive social and ethical problem that I feel the need to flag, which exists in all current Chinese language corpora used as training data for PLMs and is reflected in this paper. This is the erasure and invisibility of linguistic variability in Chinese language training data through (1) standardisation to the simplified character set where traditional characters occur in the input data, ensuring that traditional characters will not be output by the PLM, and (2) lack of acknowledgement that "Chinese" is linguistically not a single language (as recognised by the variety of ISO codes for Chinese languages in ISO 639-3, including Yue (Cantonese)) and that the written input sources may include text that has variations in word order and grammatical structure typical of, for example, Yue (e.g. Kaltenegger, S. 2020; Yan, J. 2008 (dissertation)). At a minimum, the standardisation of the character set in the underlying data used to train PLMs, and its implications for the output character set, should be explicitly flagged, as well as the geolocations from which the training data was collected. Where appropriate (i.e. if the underlying data is indeed standard Mandarin Chinese -- and the fact that most massive datasets are collected from the web makes it unlikely that the data will include only standard Mandarin Chinese), the language descriptor "Chinese" should be modified by "Mandarin" to acknowledge the existence of other varieties of Chinese.
>
> A3: Many thanks for this ethics comment. We understand your concern. As the currently used four PLMs are from public sources, we’ll contact with the developer of these models for the information of training data used for these models. We’ll try to include data cards for both the PLMs we used and our dataset itself, especially with respect to the varieties of Chinese. We conjecture that a great majority of instances of the training data are Mandarin but instances of other varieties of Chinese may be also included. Investigating the impact of the differences and varieties of Chinese on Chinese large language models would be interesting. Regarding TGEA 2.0, there are a very few generated sentences containing Yue (Cantonese) characters. During the annotation, if a sentence with Yue characters is unreadable for Mandarin-speaking annotators, such sentences are removed. We’ll provide a clear description with respect to the variety of Chinese in the next version.

---

> > ### Comment · Reviewer_7aUo · 2022-08-29
> > **Thank you for response to comments and suggestions | Ethical review response noted**
> >
> > Data cards will be an important addition to the next version of the paper. Thank you for your commitment to providing better specification of the Chinese language varieties included in the datasets in future iterations, to the extent that this is possible. It is to be hoped that researchers will take the opportunity to investigate the impact of the differences and varieties of Chinese on Chinese large language models, as you mention (the same applies to English language varieties in English LLMs, and LLMs in other languages).

---

> > > ### Author Response · Authors · 2022-08-29
> > > **Thank you very much for your great ethical comments that inspired us a lot**
> > >
> > > Thank you so much! We'll include data cards with much details on Chinese varieties in training data and take the investigation on the impact of language varieties on large language models as our important future work.

---

### Official Review · Reviewer_xbS4 · 2022-07-27
**Review of TGEA 2.0**

**Rating:** 6
**Confidence:** 4
**Correctness:** The authors properly designed annotat…
**Clarity:** The paper is well-written.

**Strengths:**

- It nicely expands previous TGEA in terms of scalability, annotation richness, etc.
- Strict quality control on the construction process
- Proposed MiSEW and pathology mitigation which can assist the annotation richness.

**Weaknesses:**

- The intention of MiSEW extraction is plausible, but is somewhat overlapped with erroneous span location as a downstream task. Thus the necessity of the task should be more justified in some manner, such as qualitative analysis.
- Proposed pathology mitigation should also be more explained in terms of why it should be jointly considered in future works.

**Additional Feedback:**

1. Minor correction
- "w/ 1 es" in Table 1: 3,1097 → 31,097

2. It would be better to discuss the future direction of class imbalance between correct/errneous texts.


**Documentation:**

It provides dataset with appropriate document form.

**Relation To Prior Work:**

It clearly compared with previous works (TGEA and ScareCrow).

**Summary And Contributions:**

This paper proposes TGEA 2.0, the largest dataset for diagnosing typed errors made by pretrained language models. It is an extended version of TGEA, with various large language models and downstream tasks. The paper mainly compared its contribution with TGEA and ScareCrow. Several experiments are performed using the dataset, and experimental results on various downstream tasks show that there are large rooms exploring the proposed dataset.

---

> ### Author Response · Authors · 2022-08-18
> **Thanks for comments and suggestions**
>
> Thank you very much for your insightful comments and suggestions. Our answers to your comments and questions are as follows.
>
>  Q1: The intention of MiSEW extraction is plausible, but is somewhat overlapped with erroneous span location as a downstream task. Thus the necessity of the task should be more justified in some manner, such as qualitative analysis.
>
> A1: As mentioned in the paper, the reasons for annotating MiSEWs are three-fold: providing justification and hence explanation to the annotated errors, reducing the amount of annotation time in writing rationales and being objective. We cannot have these advantages if only erroneous spans are annotated as they alone are usually not able to show the erroneous points. This is because errors are usually not self-evident, but associated with other words. This task has its necessity as it can evaluate the ability of models to detect and capture words associated with errors and reasoning behind errors. In addition to this, MiSEWs can be used for data augmentation to synthesize more erroneous sentences to solve the data imbalance problem. We can delete, insert or replace words in a MiSEW to generate new erroneous sentences.
>
> Q2: Proposed pathology mitigation should also be more explained in terms of why it should be jointly considered in future works.
>
> A2: Different from traditional GEC tasks, our datasets are generated from generative PLMs and one of the purposes of TGEA 2.0 is to improve the generation abilities of PLMs and to help the generative PLMs make less errors. Our dataset contains annotations, corrections, rationales of the errors exhibited in language models, providing an opportunity for language models to learn from their errors and corresponding human corrections, in addition to learning from correct texts via self-supervised learning. We’ll make this more clear in the next version. Many thanks.
>
> Q3: It would be better to discuss the future direction of class imbalance between correct/errneous texts.
>
> A3: Thanks for the valuable suggestion. We’ve discussed the imbalance issue in subsection 5.7 Discussion on Data Unbalance in the uploaded new version according to your suggestion.

---

### Official Review · Reviewer_xxwr · 2022-07-27
**Valuable large-scale pretrained language model authored text assessment data and tasks.**

**Rating:** 6
**Confidence:** 3
**Correctness:** The dataset seems to have been constr…

**Strengths:**

The main strength of this dataset is its scale: it substantially extends TGEA 1.0 to now consist of 195,629 annotated sentences. Such a dataset will be particularly useful in devising methods to assess the quality of the generated text from pre-trained language models. Also, the authors have taken great care for a sophisticated quality control process in order to ensure that the annotations for the various benchmarking tasks can be trusted.

**Weaknesses:**

Further explanation or clarification is required for the following points:

-	The authors claim that the examples generated are diverse due to the different decoding strategies and also using 4 different pretrained language models. However, are 4 pretrained language models representative in the mistakes they make for future pretrained language models that have come out recently and to come out which are far larger in size and may have different pathological weaknesses?

-	It would be great if the novelty in the tasks beyond TGEA 1.0 could be clearly spelled out beyond just the scale of the dataset.

-	How valid is it to compare the pathological weaknesses of models in Chinese to English examples as in SCARECROW?


**Additional Feedback:**

Additionally, the results section offers a large number of values for various systems. It would be good if further insights could be provided beyond simply stating the values on why certain baseline systems do better than other systems for each of the tasks.

**Clarity:**

Yes – very clear to read, but a small typo in line 32 (the grammar seems a bit off).

**Documentation:**

Yes, the baseline systems have the training regimes, hyperparameter selection and other documentation clearly provided.

**Ethics:**

No major ethical concerns. Possibly there is the risk of offensive text generated in the dataset by the pretrained language models that is distributed to the public.

**Relation To Prior Work:**

Yes, the authors outline how TGEA 2.0 builds on top of TGEA 1.0 and SCARECROW but it would be good to further highlight the differences between TGEA 2.0 and TGEA 1.0.

**Summary And Contributions:**

The authors introduce the TGEA 2.0 dataset which is a Chinese dataset where the examples are generated by various pretrained language models. The dataset has been annotated such that the machine-authored texts can be assessed on various tasks within the broad categories of diagnosis tasks and pathology mitigation tasks.

---

> ### Author Response · Authors · 2022-08-18
> **Thanks for comments and suggestions**
>
> Thank you very much for your insightful comments and suggestions. Our answers to your comments and questions are as follows.
>
>  Q1: The authors claim that the examples generated are diverse due to the different decoding strategies and also using 4 different pretrained language models. However, are 4 pretrained language models representative in the mistakes they make for future pretrained language models that have come out recently and to come out which are far larger in size and may have different pathological weaknesses?
>
> A1: The current version of the dataset covers 4 Chinese PLMs because there are only a few publicly available and reliable Chinese PLMs when we started data curation (it took over one year to collect and annotate data). We’ve tried our best to promote diversity by using language models of different scales, prompts of different types and sources. The project is long-term and still on-going. We are planning to extend the dataset to cover larger PLMs in different languages recently released, tracking the progress of language models (i.e., the changes in the types and distributions of pathological weaknesses) in text generation.
>
> Q2: It would be great if the novelty in the tasks beyond TGEA 1.0 could be clearly spelled out beyond just the scale of the dataset.
>
> A2: There are four key differences between TGEA 2.0 and TGEA 1.0 while the scale difference is only one of them. First, we use multiple models, prompts of different types and sources, decoding strategies to generate texts while TGEA 1.0 only uses a single model with nominal prompts. Second, we define MiSEW to facilitate human annotation and MiSEW Extraction as a new task. Third, we perform Error Type classification and Error correction based on ground-truth erroneous spans to measure the degree of difficulties of these tasks as standalone tasks. Finally, we propose Generation Pathology Mitigation tasks for generative PLMs, which has been not defined at all in TGEA 1.0. We’ll make the differences from TGEA 1.0 more clear in the next version.
>
> Q3: How valid is it to compare the pathological weaknesses of models in Chinese to English examples as in SCARECROW?
>
>  A3: The comparison to SCARECROW is only to show the common problems exhibited in both Chinese and English text generation by language models. Due to the differences in error taxonomy, languages, language models used, annotation procedures, etc., the comparison is not a direct comparison. We’d like to compare different languages with the same model architectures trained with the same amount of data and data domain, the same annotation scheme, etc., in the future.
>
>  Q4: Additionally, the results section offers a large number of values for various systems. It would be good if further insights could be provided beyond simply stating the values on why certain baseline systems do better than other systems for each of the tasks.
>
>  A4: Thanks for the valuable suggestion. We’ve added a new subsection 5.7 Discussion on Data Unbalance and 5.8 Analysis on Generation Pathology Mitigation for the reasons behind the results.

---

### Official Review · Reviewer_f3qm · 2022-07-28
**Well-constructed Dataset for Language Models**

**Rating:** 7
**Confidence:** 4

**Strengths:**

1. Data collection phase is well-implemented: The authors use different models, decoding strategies and prompt types (nominal, phrasal and sentential) with various domains (News, Wikipedia and Web Fictions) to diversify types of erroneous sentences.
2. The annotation process and quality control are well-designed to annotate the large-scale dataset while maintaining its quality.
3. The dataset has potential usages: Large language models in Chinese can benefit from training on this dataset to mitigating erroneous sentences. Also, the discriminative models can be trained to automatically detect errors made by language models. Thus, this work can be valuable for future research.

**Weaknesses:**

1. I have some concerns regarding the quality control:
* L188: Who trained the first 4 reviewers? I would like to evaluate the quality of this dataset carefully, since their annotations are used as ground-truths to train other annotators.
* L199: What is average performance of 7 well-trained reviewers? Are they trained by the annotations produced by the first 4 reviewers? The reason I asked is that they are the ones who guarantee the high-quality outcomes for this dataset.
2. Although alpha-balanced loss was used, some diagnostic tasks such as Erroneous Text Detection, MiSEW Extraction suffer a heavy unbalance that may affect model training and evaluation, so the results on these datasets are not quite convincing.
3. No statistics reported for the proposed tasks in two sets of benchmarks.
4. The Word Prediction task in the pathology mitigation benchmark does not properly evaluate the ability of language models because there can be many correct predictions for the last token given each sentence.
5. No qualitative examples (i.e., model predictions) for each benchmark task in the main text and the appendix.
6. Some minor issues in presentation:
* Numbers in Table 1 are quite small that makes it hard to read
* L240: Error Correction task is missing.
* L267: It should be MacBERT instead of MacBEERT.

**Additional Feedback:**

There are two suggestions for improvements:
1. Provide human performance to see the upperbound of proposed benchmark tasks.
2. Train the other 3 models used in data collection phase (i.e., NEZHA-Gen, PanGu-alpha, CPM) and ask annotators/reviewers to evaluate their generated texts to see if training on the dataset helps them avoid making errors.

and two questions regarding data collection:
1. L148: How to obtain 20K paragraphs from 170K prompts? Did you filter out any unqualified paragraphs?
2. L151: With only the first sentence selected from 20K paragraphs (thus, 20K sentences in total), I am not quite sure how to obtain 195,629 annotated sentences.

**Clarity:**

Although there are some minor issues (see weaknesses), the paper is quite clear and well written in general.

**Correctness:**

Overall, the dataset is well-constructed given the carefully designed data collection and annotation processes, especially the quality control. The proposed benchmarks derived from the dataset have some flaws and need to be improved for better evaluation of language models.

**Documentation:**

Documentation of this paper is detailed and quite clear, but can be improved.

**Ethics:**

There is no ethical issue.

**Relation To Prior Work:**

The paper provides a detailed discussion about the prior work and also points out the differences so I think they have a solid contribution.

**Summary And Contributions:**

This paper presents a large-scale and curated dataset in Chinese along with two benchmarks for diagnosis (5 tasks) and pathology mitigation (2 tasks) to improve quality of generated texts from language models. The authors designed a thorough annotation process including data collection, training annotators in the pre-annotation phase and quality control by the feedback loop. The selected sentences for annotation cover 3 aspects: model, decoding strategy and prompt. They also provided a detailed analysis on the distributions of erroneous sentences produced by a variety of models with different sizes ranging from 110M to 2.6B parameters. The experimental results on the proposed benchmarks show that the diagnosis tasks are challenging and training a language model (here, GPT-2) on their dataset help reducing errors in the generated texts.

---

> ### Author Response · Authors · 2022-08-18
> **Thanks for comments and suggestions**
>
> Thank you very much for your insightful comments and suggestions. Our answers to your comments and questions are as follows.
>
> Q1: Who trained the first 4 reviewers? I would like to evaluate the quality of this dataset carefully, since their annotations are used as ground-truths to train other annotators. What is average performance of 7 well-trained reviewers? Are they trained by the annotations produced by the first 4 reviewers? The reason I asked is that they are the ones who guarantee the high-quality outcomes for this dataset.
>
> A1: Two reviewers in TGEA 1.0 [1] with an average IAA of 92.3% and Cohen’s Kappa of 82.6% trained the first 4 reviewers and 7 well-trained reviewers. All these reviewers were trained in multiple rounds with annotation conventions. And the average Cohen's Kappa statistics of 7 reviewers is 56.6%.
>
> Q2: Although alpha-balanced loss was used, some diagnostic tasks such as Erroneous Text Detection, MiSEW Extraction suffer a heavy unbalance that may affect model training and evaluation, so the results on these datasets are not quite convincing.
>
> A2:  The results on some tasks, e.g., Erroneous text detection, MiSEW Extraction, suffer a heavy unbalance even alpha-balanced loss was used. We speculate that the reasons for this are twofold. First, erroneous sentences chosen from those generated by CLMs are also challenging for MLMs (e.g., for them to detect MiSEWs, erroneous spans). Second, the differences (errors) between an erroneous and human-corrected text are usually only a few characters while the average number of characters in erroneous sentences is large (over 40 characters). These errors can only be detected and corrected with long-distance dependencies. Data augmentation methods can be used on TGEA 2.0 to balance the ratio of correct and erroneous sentences. Specifically, we can expand the number of erroneous sentences based on MiSEW or using a trained Sequence-to-Sequence model to convert generated texts into erroneous texts. We leave this to our future work. We’ve also added the above analysis to subsection 5.7 in the uploaded new version.
>
> Q3: No statistics reported for the proposed tasks in two sets of benchmarks.
>
> A3: Thanks for the valuable suggestion. We’ve now reported statistics in Appendix A.2 Table 4. For your ease of reference, we've also provided the table below.
>
> | Tasks                     |  Train  |  Dev   |  Test  |
> | ------------------------- | :-----: | :----: | :----: |
> | Erroneous Text Detection  | 156,502 | 19,563 | 19,564 |
> | MiSEW Extraction          | 28,818  | 3,602  | 3,601  |
> | Erroneous Span Location   | 28,818  | 3,602  | 3,601  |
> | Error Type Classification | 33,666  | 4,181  | 4,220  |
> | Error Correction          | 28,818  | 3,602  | 3,601  |
> | Pairwise Comparision      | 156,502 | 3,602  | 3,601  |
> | Word Prediction           | 156,502 | 3,024  | 3,025  |
>
> Q4: The Word Prediction task in the pathology mitigation benchmark does not properly evaluate the ability of language models because there can be many correct predictions for the last token given each sentence.
>
> A4: The word prediction task predicts the target token according to prefix, evaluating the ability of modeling and capturing long-distance dependencies of language models in text generation. The main reasons of using this task are that it is model-agnostic and can be easily automatically evaluated. To take multiple predictions into account, we also consider top-3 accuracy. Evaluating the ability of language models with automatic metrics itself is very challenging. We’ll define better evaluation methods in the future.
> Q5: No qualitative examples (i.e., model predictions) for each benchmark task in the main text and the appendix.
>
> A5: Thanks for the valuable suggestion. We’ve added examples of model predictions in Appendix Table 9.
>
> Q6: How to obtain 20K paragraphs from 170K prompts? Did you filter out any unqualified paragraphs? With only the first sentence selected from 20K paragraphs (thus, 20K sentences in total), I am not quite sure how to obtain 195,629 annotated sentences.
>
> A6: 10k prompts were shared by the four PLMs while the remaining 160k were divided equally across the four models, so the total number of prompts used to generate paragraphs are 200k (160k+4X10k). We filtered unqualified paragraphs (via both automatic and manual check & review) from 200k, resulting in 195,629 sentences.

---

> > ### Author Response · Authors · 2022-08-18
> > **Thanks for comments and suggestions**
> >
> > Q7: Provide human performance to see the upperbound of proposed benchmark tasks.
> >
> > A7:  Thanks for this suggestion. We provide human performance for two tasks, Erroneous Text Detection and Pairwise Comparision, in the table below. For other tasks, we'll provide the results in the next version.
> >
> > | Erroneous Text Detection | Accuracy (%) | Precision (%) | Recall (%) | F1 (%) |
> > | ------------------------ | :----------: | :-----------: | :--------: | :----: |
> > | Dev                      |    91.13     |     76.19     |   72.72    | 74.42  |
> > | Test                     |    91.39     |     73.08     |   76.00    | 74.51  |
> >
> > | Pairwise Comparision | Accuracy (%) |
> > | -------------------- | :----------: |
> > | Dev                  |    87.32     |
> > | Test                 |    86.79     |
> >
> > Q8: Train the other 3 models used in data collection phase (i.e., NEZHA-Gen, PanGu-alpha, CPM) and ask annotators/reviewers to evaluate their generated texts to see if training on the dataset helps them avoid making errors.
> >
> > A8:  Thanks for this suggestion. We’ve trained two larger PLMs PanGu-alpha (2.6B) and CPM (2.6B) used in the data collection phase with mixed-tuning. We randomly selected 100 prompts that cause models to generate erroneous sentences for each trained PLM. Then we used the best decoding strategies as mentioned in Section 3.1 for models to generate sentences and asked reviewers to evaluate their generated texts. We find that the percentage of erroneous sentences of PanGu-alpha/CPM drops from 22.1%/17.62% to 14%/13%.
> >
> > [1] Jie He, Bo Peng, Yi Liao, Qun Liu and Deyi Xiong. TGEA: An Error-Annotated Dataset and Benchmark Tasks for TextGeneration from Pretrained Language Models. ACL 2021.

---

> ### Comment · Reviewer_f3qm · 2022-08-27
> **Questions addressed by the authors**
>
> Thank you very much for your efforts in providing human performance and also more interesting experimental results!
>
> * **Clarification**: The points that were not clear to me in the Data Construction and Experiments sections were well-explained.
>
> * **Paper writing improved**: For a dataset paper, we need to provide the statistics and qualitative examples for readers to gain more insights about the proposed dataset. The authors have updated the paper adding such information and also provided human performance that is appreciated.
>
> * **Drawbacks acknowledgment**: Although there are still some drawbacks of this paper that are not yet addressed, such as the proposed tasks suffer heavy unbalance or the automatic metrics for evaluating the ability of language models, they are acknowledged by the authors and should be improved in the future work.
>
> * **Dataset value**: The large-scale dataset and benchmark tasks are useful to evaluate and improve Chinese language models in terms of reducing erroneous-generated sentences (proof is also provided by the authors in the answer A8). The tasks are also challenging since there is a big gap between CLMs and human performance (as shown in the answer A7), that would lead to better language models in the future.
>
> For the above reasons, I increase my score from 6 to 7.

---

> > ### Author Response · Authors · 2022-08-29
> > **Many thanks for your great suggestions and questions**
> >
> > Thank you so much for your great suggestions and questions. We'll continue to improve the writing of our paper and provide evaluation results for the texts generated by the models that are used in the data collection phase and then fine-tuned on our dataset as you suggest in the next version.

---

### Review · Ethics_Reviewer_DVTQ · 2022-08-22

**Recommendation:** 2

**Ethics Documentation:**

To address the reviewers' concerns on erasure of specificities in the Chinese language, the authors offered to contact the developers of the publicly available models they are using with the idea of asking for information on the training data used for these models.
The authors also offered to include data cards for th emodels and datasets, especially with respect to the varieties of Chinese. The goal is to identify varieties of Chinese other than Mandarin.
More generally, authors propose to provide a clear description with respect to the variety of Chinese in the revised version of the paper.


**Ethics Review:**

The authors introduce the TGEA 2.0 dataset which is a Chinese dataset where the examples are generated by various pretrained language models. The dataset has been annotated such that the machine-authored texts can be assessed on various tasks within the broad categories of diagnosis tasks and pathology mitigation tasks.

The main issue raised by reviewers is the risk of erasure and invisibility of linguistic variability in Chinese language training data. A recommendation was formulated in this regard.

---

> ### Author Response · Authors · 2022-08-27
> **Many thanks for the Great Suggestion on the Linguistic Variability in Chinese Training Data**
>
> Thank you very much for the great suggestion.
>
> We've manually checked TGEA 2.0 and found that more than 99% texts are simplified Chinese. However, some machine-authored texts are traditional Chinese. We provide traditional Chinese examples generated by each model below. We conjecture that sources for the training data of the used 4 PLMs contain traditional Chinese texts, which are also roughly shown below. We'll provide more details on this in the revised version.
>
> ### GPT2
>
> **Model Location**: https://github.com/ghosthamlet/gpt2-ml-torch
>
> **Data Sources**: THUCNews (http://thuctc.thunlp.org/#%E4%B8%AD%E6%96%87%E6%96%87%E6%9C%AC%E5%88%86%E7%B1%BB%E6%95%B0%E6%8D%AE%E9%9B%86THUCNews), Nlp_chinese_corpus (https://github.com/brightmart/nlp_chinese_corpus)
>
> **Varieties of Chinese**:  Simplified / Traditional Chinese
>
> **Traditional Chinese Examples in TGEA 2.0**: 這些人大部份都從小就被送去國外接受高等教育，所以學到了非常先進的知和思想，加上身一群有見解、有智慧、會播知的人，自然會成為佼佼者。/ Most of these people have been sent abroad for higher education when they were children , so they have learned very advanced knowledge and ideas, and they will naturally become outstanding as a group of people with insight, wisdom and knowledge.
>
> ### CPM
>
> **Model Location**: https://github.com/TsinghuaAI/CPM-1-Generate
>
> **Data Sources**: WuDaoCorpus (https://data.wudaoai.cn/)
>
> **Varieties of Chinese**:  Simplified / Traditional Chinese
>
> **Traditional Chinese Examples in TGEA 2.0**: 很多人也不知道怎麼去想這个問題,為什麼就會有這樣的結論呢?首先先給自己找個理由吧! / Many people don't know how to think about this issue. Why do they come to such a conclusion? First, find a reason for yourself!
>
> ### NEZHA
>
> **Model Location**: https://github.com/huawei-noah/Pretrained-Language-Model/tree/master/NEZHA-Gen-TensorFlow
>
> **Data Sources**: Chinese Wikipedia:  https://zh.wikipedia.org/wiki/, Baidu Baike (https://baike.baidu.com/), Chinese News: multiple news websites(e.g. Sina News https://news.sina.com.cn/).
>
> **Varieties of Chinese**:  Simplified / Traditional Chinese
>
> **Traditional Chinese Examples in TGEA 2.0**: 所以,爲了保持流動性不會過剩，必须要把流动性進行適當調節。 / Therefore, in order to keep the liquidity from being excessive, the liquidity must be properly adjusted.
>
> ### PanGu
>
> **Model Location**: https://github.com/huawei-noah/Pretrained-Language-Model/tree/master/PanGu-%CE%B1
>
> **Data Sources**:  Public datasets: DuReader (https://github.com/baidu/DuReader)：CAIL2018 (https://github.com/china-ai-law-challenge/CAIL2018), Sogou-CA (http://www.sogou.com/labs/resource/ca.php) etc. Encyclopedia: Baidu Baike (https://baike.baidu.com/),  Sogou Baike (https://baike.sogou.com/v62768613.htm?ch=zhihu.topic), etc. E-books: not available (novels, history, poetry, ancient prose). Common Crawl (https://commoncrawl.org/the-data/). news: not available.
>
> **Varieties of Chinese**:  Simplified / Traditional Chinese
>
> **Traditional Chinese Examples in TGEA 2.0**: 每個區域都有明確的表格,讓觀眾可以清楚知道自己是否被限制在某個範圍內;而每部劇作也均會根據不同的要求對表演進行調整,因此觀眾能夠自由地觀看和選擇任何一部作品。 / Each area has a clear table, so that the audience can clearly know whether they are confined to a certain range; and each play will also adjust the performance according to different requirements, so the audience is free to watch and choose any one works.

---

### Official Review · Reviewer_qfey · 2022-08-22
**This is not a review. My ethics review has been done for this paper.**

**Rating:** 7
**Confidence:** 4
**Correctness:** N/A
**Clarity:** N/A

**Strengths:**

To address the reviewers' concerns on erasure of specificities in the Chinese language, the authors offered to contact the developers of the publicly available models they are using with the idea of asking for information on the training data used for these models. The authors also offered to include data cards for th emodels and datasets, especially with respect to the varieties of Chinese. The goal is to identify varieties of Chinese other than Mandarin. More generally, authors propose to provide a clear description with respect to the variety of Chinese in the revised version of the paper.


**Weaknesses:**

N/A

**Additional Feedback:**

N/A

**Documentation:**

N/A

**Ethics:**

See ethics review.

**Relation To Prior Work:**

N/A

**Summary And Contributions:**

Recommendation: 2: Serious ethical issues that need to be addressed in the final version

Ethics Review:

The authors introduce the TGEA 2.0 dataset which is a Chinese dataset where the examples are generated by various pretrained language models. The dataset has been annotated such that the machine-authored texts can be assessed on various tasks within the broad categories of diagnosis tasks and pathology mitigation tasks.

The main issue raised by reviewers is the risk of erasure and invisibility of linguistic variability in Chinese language training data. A recommendation was formulated in this regard.
Ethics Documentation:

---

### Meta-Review · Area_Chair_UJ3P · 2022-09-09

**Recommendation:** Accept
**Confidence:** 4

**Metareview:**

The reviewers all liked the paper. The authors' response clarified most points raised by the reviewers. In view of that, the authors are strongly invited to take the feedback on board for the final version. The main ethical issue raised by reviewers is the risk of erasure and invisibility of linguistic variability in Chinese language training data. Data cards need to be added to the final version.

---

### Decision · Program_Chairs · 2022-09-16

Accept